# Oncogenic hijacking of a developmental transcription factor evokes vulnerability toward oxidative stress in Ewing sarcoma

Aruna Marchetto [1], Shunya Ohmura [1], Martin F. Orth [1], Maximilian M. L. Knott [1,2], Maria V. Colombo [3], Chiara Arrigoni [3], Victor Bardinet[4], David Saucier [5], Fabienne S. Wehweck[1,2], Jing Li[1], Stefanie Stein[1], Julia S. Gerke[1], Michaela C. Baldauf[1], Julian Musa [1,6], Marlene Dallmayer[1], Laura Romero-Pérez [1,7,8], Tilman L. B. Hölting [1], James F. Amatruda [5,9], Andrea Cossarizza [10], Anton G. Henssen [4,11,12,13,14], Thomas Kirchner[2,14,15], Matteo Moretti[3], Florencia Cidre-Aranaz [1,7,8], Giuseppina Sannino[1] & Thomas G. P. Grünewald [1,2,7,8,15,16 ✉]

Ewing sarcoma (EwS) is an aggressive childhood cancer likely originating from mesenchymal stem cells or osteo-chondrogenic progenitors. It is characterized by fusion oncoproteins involving EWSR1 and variable members of the ETS-family of transcription factors (in 85% FLI1). EWSR1-FLI1 can induce target genes by using GGAA-microsatellites as enhancers. Here, we show that EWSR1-FLI1 hijacks the developmental transcription factor SOX6 – a physiological driver of proliferation of osteo-chondrogenic progenitors – by binding to an intronic GGAA-microsatellite, which promotes EwS growth in vitro and in vivo. Through integration of transcriptome-profiling, published drug-screening data, and functional in vitro and in vivo experiments including 3D and PDX models, we discover that constitutively high SOX6 expression promotes elevated levels of oxidative stress that create a therapeutic vulnerability toward the oxidative stress-inducing drug Elesclomol.

Collectively, our results exemplify how aberrant activation of a developmental transcription factor by a dominant oncogene can promote malignancy, but provide opportunities for targeted therapy.

[1] Max-Eder Research Group for Pediatric Sarcoma Biology, Institute of Pathology, Faculty of Medicine, LMU Munich, Munich, Germany. [2] Institute of Pathology, Faculty of Medicine, LMU Munich, Munich, Germany. [3] Regenerative Medicine Technologies Laboratory, Ente Ospedaliero Cantonale (EOC), Lugano, Switzerland. [4] Department of Pediatrics, Division of Oncology and Hematology, Charité Berlin, Berlin, Germany. [5] Department of Pediatrics and Molecular Biology, University of Texas Southwestern Medical Center and Children's Medical Center, Dallas, TX, USA. [6] Department of General, Visceral and Transplantation Surgery, University of Heidelberg, Heidelberg, Germany. [7] Hopp-Children's Cancer Center (KiTZ), Heidelberg, Germany. [8] Division of Translational Pediatric Sarcoma Research, German Cancer Research Center (DKFZ), Heidelberg, Germany. [9] Children's Hospital of Los Angeles, Los Angeles, CA, USA. [10] Department of Medical and Surgical Sciences for Children and Adults, University of Modena and Reggio Emilia School of Medicine, Modena, Italy. [11] Berlin Institute of Health, Berlin, Germany. [12] Experimental and Clinical Research Center (ECRC) of the MDC and Charité Berlin, Berlin, Germany. [13] German Cancer Consortium (DKTK), partner site, Berlin, Germany. [14] German Cancer Research Center (DKFZ), Heidelberg, Germany. [15] German Cancer Consortium (DKTK), partner site, Munich, Germany. [16] Institute of Pathology, University Hospital Heidelberg, Heidelberg, Germany. ✉email: t.gruenewald@dkfz-heidelberg.de

Ewing sarcoma (EwS) is the second most common bone or soft-tissue cancer in children and adolescents[1]. Even though the cell of origin of EwS is still debated, increasing evidence suggests that it may arise from mesenchymal stem cell (MSC)-derived early committed osteo-chondrogenic progenitors[2,3]. Indeed, EwS cells display a highly undifferentiated and embryonal phenotype. Clinically, EwS is a rapidly metastasizing cancer, and ~25% of cases are metastatic at initial diagnosis[1]. While great advances in treatment of localized disease have been achieved, established therapies still have limited success in advanced stages despite high toxicity[1]. Thus, more specific and in particular less toxic therapies are urgently required.

As a genetic hallmark, EwS tumors express chimeric EWSR1-ETS (EwS breakpoint region 1—E26 transformation specific) fusion oncoproteins generated through fusion of the EWSR1 gene and variable members of the ETS-family of transcription factors, most commonly FLI1 (85% of all cases)[4,5]. Prior studies demonstrated that EWSR1-FLI1 acts as a pioneer transcription factor that massively rewires the tumor transcriptome ultimately promoting the malignant phenotype of EwS[6,7]. This is in part mediated through interference with and/or aberrant activation of developmental pathways[3,8]. Remarkably, EWSR1-FLI1 regulates ~40% of its target genes by binding to otherwise non-functional GGAA-microsatellites (mSats)[9] that are thereby converted into potent de novo enhancers, whose activity increases with the number of consecutive GGAA-repeats[7,10–12].

Although EWSR1-FLI1 would in principle constitute a highly specific target for therapy, this fusion oncoprotein proved to be notoriously difficult to target due to its intranuclear localization, its activity as a transcription factor[13,14], the absence of regulatory protein residues[1], its low immunogenicity[15], and the high and ubiquitous expression of its constituting genes in adult tissues[1]. Hence, we reasoned that developmental genes and pathways that are aberrantly activated by EWSR1-FLI1 and virtually inactive in normal adult tissues, could constitute druggable surrogate targets.

As EwS most commonly arises in bone and possibly descends from osteo-chondrogenic progenitor cells[3], we speculated that EWSR1-FLI1 might interfere with bone developmental pathways. The transcription and splicing factor SOX6 (SRY-box 6) plays an important role in endochondral ossification[16]. Interestingly, its transient high expression delineates cells along the osteo-chondrogenic lineage showing high rates of proliferation while maintaining an immature phenotype along this lineage[17–19].

In the current study, we show that EWSR1-FLI1 binds to an intronic GGAA-mSat within SOX6, which acts as an EWSR1-FLI1-dependent enhancer that induces the high and constitutive overexpression of SOX6 in EwS tumors. Moreover, we report that SOX6 promotes proliferation and tumorigenicity of EwS cells, and confers a druggable therapeutic vulnerability toward the oxidative stress-inducing small-molecule Elesclomol, through upregulation of cell intrinsic oxidative stress starting from mitochondria.

## Results

### SOX6 is highly but variably expressed in EwS.
To explore the expression pattern of SOX6, we took advantage of a well-curated set of >750 DNA microarrays[20], comprising 18 representative normal tissues types and 10 cancer entities. Comparative analyses revealed that SOX6 is overexpressed in EwS relative to normal tissues and other cancers (Fig. 1a). These data were validated at the protein level in a comprehensive tissue microarray[20], comprising the same normal tissue types and cancer entities (Fig. 1b, c). Both analyses showed that SOX6 is highly expressed in EwS tumors, albeit with substantial inter-tumor heterogeneity.

The generally high but variable expression of SOX6 was also observed in EwS cell line models compared to cell lines of three other pediatric cancer types including osteosarcoma (U2OS and SAOS-2), neuroblastoma (TGW and SK-N-AS) and rhabdomyosarcoma (Rh36 and Rh4) (Supplementary Fig. 1).

### EWSR1-FLI1 induces SOX6 expression via an intronic GGAA-mSat.
The relatively high expression of SOX6 in EwS compared to other sarcomas and pediatric cancers implied that there might be a regulatory relationship with the EwS specific fusion oncogene EWSR1-FLI1. Indeed, knockdown of EWSR1-FLI1 in A673/TR/shEF1 and SK-N-MC/TR/shEF1 cells harboring a doxycycline (Dox)-inducible short hairpin RNA (shRNA) against the fusion gene, strongly reduced SOX6 expression in a time-dependent manner in vitro (Fig. 2a, Supplementary Fig. 2a) and in vivo (Fig. 2b). Conversely, ectopic expression of EWSR1-FLI1 in human embryoid bodies strongly induced SOX6 expression (Fig. 2c).

To investigate the underlying mechanism of this regulatory relationship, we analyzed publicly available DNase-Seq and ChIP-Seq data of two EwS cell lines (A673 and SK-N-MC) and found a prominent EWSR1-FLI1 peak within intron 1 of SOX6, which was strongly reduced upon EWSR1-FLI1 knockdown (Fig. 2d). This EWSR1-FLI1 peak mapped to a GGAA-mSat located within a DNase 1 hypersensitivity site, indicating open chromatin, and showed EWSR1-FLI1-dependent acetylation of H3K27, which marks active enhancers (Fig. 2d). The EWSR1-FLI1-dependent enhancer activity of this GGAA-mSat was confirmed by luciferase reporter assays in A673/TR/shEF1 cells transfected with pGL3 reporter plasmids in which we cloned a 1-kb fragment containing this SOX6-associated GGAA-mSat from the human reference genome (Fig. 2e).

As prior studies showed that the enhancer activity at EWSR1-FLI1-bound GGAA-mSats positively correlates with the number of consecutive GGAA-repeats[10,21,22], we hypothesized that the observed variability in SOX6 expression might be caused by differences in repeat numbers at the SOX6-associated GGAA-mSat. To test this possibility, we cloned both alleles of this mSat from eight EwS cell lines with largely different SOX6 expression levels (Supplementary Fig. 1), determined their repeat number by Sanger sequencing, and measured their enhancer activity in reporter assays. We observed a statistically significant ($P = 0.016$) positive correlation of the average SOX6 expression levels with the relative average enhancer activity across cell lines, which corresponded to the average repeat numbers of both alleles (Fig. 2f, Supplementary Table 1). Interestingly, the inter-individual differences in SOX6 expression levels correlated neither with (minor) differences of SOX6 promoter methylation nor with copy number variations at the SOX6 locus in primary EwS tumors (Supplementary Fig. 2b, c).

Collectively, these data suggest that EWSR1-FLI1 induces SOX6 by binding to a polymorphic intronic GGAA-mSat, which exhibits length-dependent enhancer activity.

### SOX6 promotes proliferation of EwS in vitro and in vivo.
To investigate the possible function of SOX6 in EwS, we generated two cell lines (RDES and TC-32) with Dox-inducible shRNAs against SOX6 (shSOX6_2 and shSOX6_3) and corresponding controls with a Dox-inducible non-targeting control shRNA (shCtrl). In these transduced cells, addition of Dox (0.1 µg/ml) to the culture medium effectively silenced SOX6 expression at the mRNA and protein level (Fig. 3a, Supplementary Fig. 3a).

Since SOX6 acts—depending on the cellular context—as a splicing and/or transcription factor[23,24], we examined the effect of SOX6 knockdown in RDES and TC-32 EwS cell lines using Affymetrix Clariom D arrays, which enable the simultaneous transcriptome-wide analysis of splicing events and differential

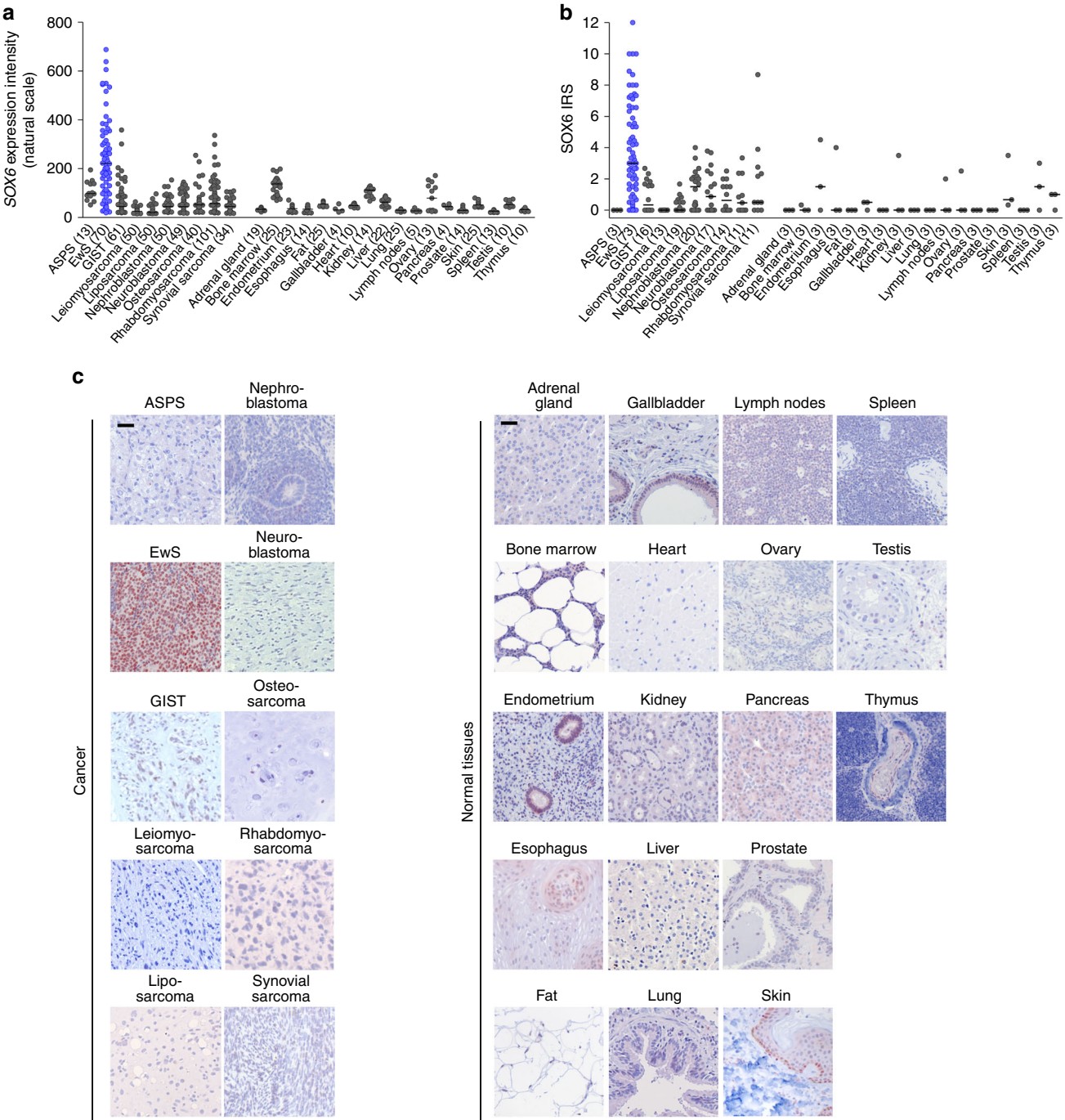

**Fig. 1 SOX6 is highly but variably expressed in EwS. a** *SOX6* expression levels (Affymetrix microarrays) in EwS tumors, nine additional sarcoma or pediatric tumor entities, and 18 normal tissue types. Data are represented as dot plots, horizontal bars represent medians. The number of biologically independent samples per group (*n*) is given in parentheses. ASPS, alveolar soft part sarcoma; GIST, gastrointestinal stromal tumor. **b** Validation of SOX6 protein expression by IHC in the same tissue types as shown in (**a**). Immuno Reactive scores (IRS) are presented as dot plots. Horizontal bars represent medians. The number of biologically independent samples per group (*n*) is given in parentheses. **c** Representative micrographs of the IHC stains from samples of the same tumor entities or normal tissue types indicated in (**b**); scale bars = 20 µm. Source data are provided as a Source Data file.

gene expression. While the knockdown of *SOX6* for 96 h had little effect on splicing (Supplementary Data 1), we noted a strong effect on differential gene expression (Fig. 3b). In fact, *SOX6* silencing induced a concordant up- or downregulation (min. absolute log2 FC of 1) of 54 and 499 genes, respectively, across shRNAs and cell lines (Supplementary Data 2). Gene set enrichment analysis (GSEA) of the entire set of differentially expressed genes (DEGs) identified a strong depletion of

proliferation-related gene signatures in SOX6-silenced EwS cells (Fig. 3b, Supplementary Data 3).

To validate the predicted role of SOX6 in EwS proliferation, we performed knockdown experiments using pooled short interfering RNAs (sipool) against *SOX6* in five EwS cell lines (A673, EW7, POE, RDES, and TC-32). Each sipool consisted of 30 different siRNAs, which virtually eliminates off-target effects[25], and which induced a 60–80% *SOX6* knockdown as compared to a

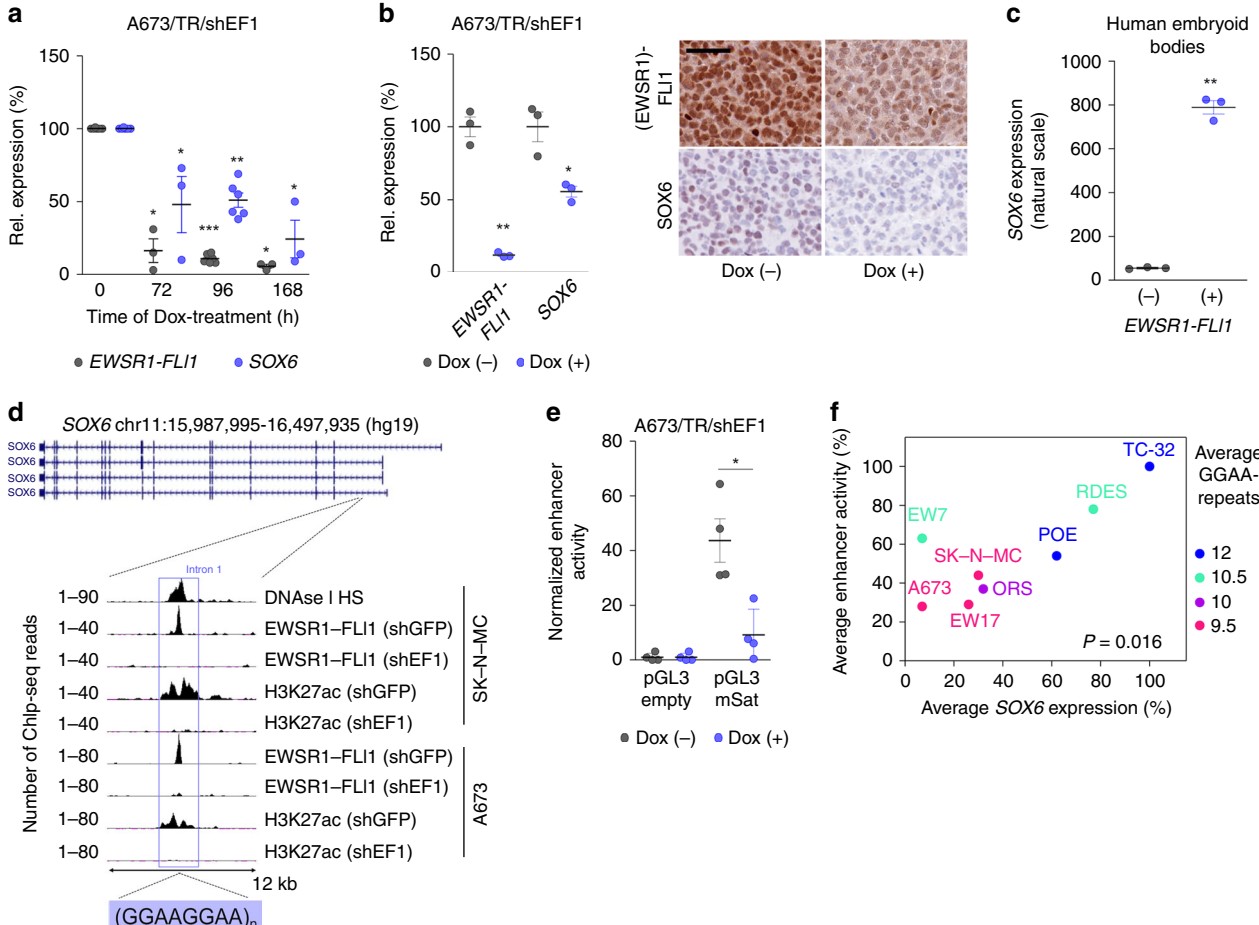

**Fig. 2 EWSR1-FLI1 induces *SOX6* expression via an intronic GGAA-mSat. a** *EWSR1-FLI1* and *SOX6* expression (qRT-PCR) in A673/TR/shEF1 cells after addition of Dox. Horizontal bars represent means, $n = 3$ biologically independent experiments for time points 0 h, 72 h and 168 h, $n = 6$ biologically independent experiments for the time point 96 h. *P* values determined by two-sided Mann–Whitney test. *EWSR1-FLI1*: 72 h ($P = 0.012$), 96 h ($P = 0.0007$), 168 h ($P = 0.012$); *SOX6*: 72 h ($P = 0.017$), 96 h ($P = 0.001$), 168 h ($P = 0.017$). **b** Left: *EWSR1-FLI1* and *SOX6* expression (Affymetrix microarrays) in A673/TR/shEF1 xenografts after 96 h of Dox-treatment. Horizontal bars represent means, $n = 3$ biologically independent xenografts per group. *P* value determined via two-sided independent one-sample *t*-test (*EWSR1-FLI1*: $P = 0.002$, *SOX6* $P = 0.028$). Right: Representative immunohistological staining for (EWSR1-)FLI1 and SOX6. Scale bar = 20 μm. **c** *SOX6* expression (Affymetrix microarrays) in embryoid bodies after ectopic *EWSR1-FLI1* expression. Horizontal bars represent means, $n = 3$ biologically independent samples per group. *P* value determined via unpaired two-sided *t*-test with Welch's correction ($P = 0.002$). **d** Integrated genomic view of DNAse I hypersensitivity (HS) and ChIP-Seq data for EWSR1-Fli1 and H3K27ac in EwS cells transfected with shRNA against EWSR1-FLI1 (shEF1) or control shRNA (shGFP). **e** Relative enhancer activity of the *SOX6*-associated GGAA-mSat in A673/TR/shEF1 cells (−/+ Dox). Horizontal bars represent means, $n = 4$ biologically independent experiments. *P* value determined via two-sided Mann–Whitney test ($P = 0.029$). **f** Correlation of the average enhancer activity of both alleles of the *SOX6*-associated GGAA-mSat and the average *SOX6* levels across eight EwS cell lines (TC-32 set as reference). The color code indicates the average number of consecutive GGAA-repeats of both alleles. *P* value determined via two-tailed Pearson correlation test, $n = 4$ biologically independent experiments. All error bars represent SEM. ***$P < 0.001$, **$P < 0.01$, *$P < 0.05$. Source data are provided as a Source Data file.

non-targeting control sipool (sipCtrl) after 96 h. In these experiments, we noted a significant reduction of the viable cell count in all EwS cell lines in standardized cell counting experiments (including the supernatant) (Supplementary Fig. 3b). Interestingly, sipool-mediated silencing of *SOX6* had no effect on proliferation of A673 and EW7 cell lines, which only express low baseline levels of *SOX6* (Supplementary Fig. 1a, Supplementary Fig. 3b). In accordance, Dox-induced long-term *SOX6* knockdown significantly reduced the 2D clonogenic and 3D sphere-formation capacities of the highly *SOX6* expressing EwS cell lines RDES and TC-32 as compared to controls (Dox (−) and shCtrl) (Fig. 3c, Supplementary Fig. 3c), while *SOX6* silencing in the *SOX6*-low expressing cell line A673 had no effect on sphere-formation capacity (Fig. 3c). To test whether this effect was mediated via alteration of the cell cycle, we carried out flow

cytometric assays with propidium iodide (PI). In serum-starved and thus $G_0$-synchronized cells, we observed a significant delay in cell cycle progression 20 h after re-addition of serum in *SOX6*-silenced cells (Supplementary Fig. 3d).

Since ~85% of all clinical EwS tumors arise in bone[1], we examined the effect of *SOX6* silencing in a 3D in vitro mineralized bone model containing viable human osteoblasts[26,27]. Similar to our previous in vitro assays, knockdown of *SOX6* reduced growth of EwS cells in this 3D bone model (Supplementary Fig. 3e). Interestingly, this effect was accompanied by a reduced expression of the proliferation marker Ki67 (Supplementary Fig. 3e), which further supported the involvement of SOX6 in cell cycle progression.

To assess the potential contribution of SOX6 to EwS tumor growth in vivo, we performed xenograft experiments by injecting

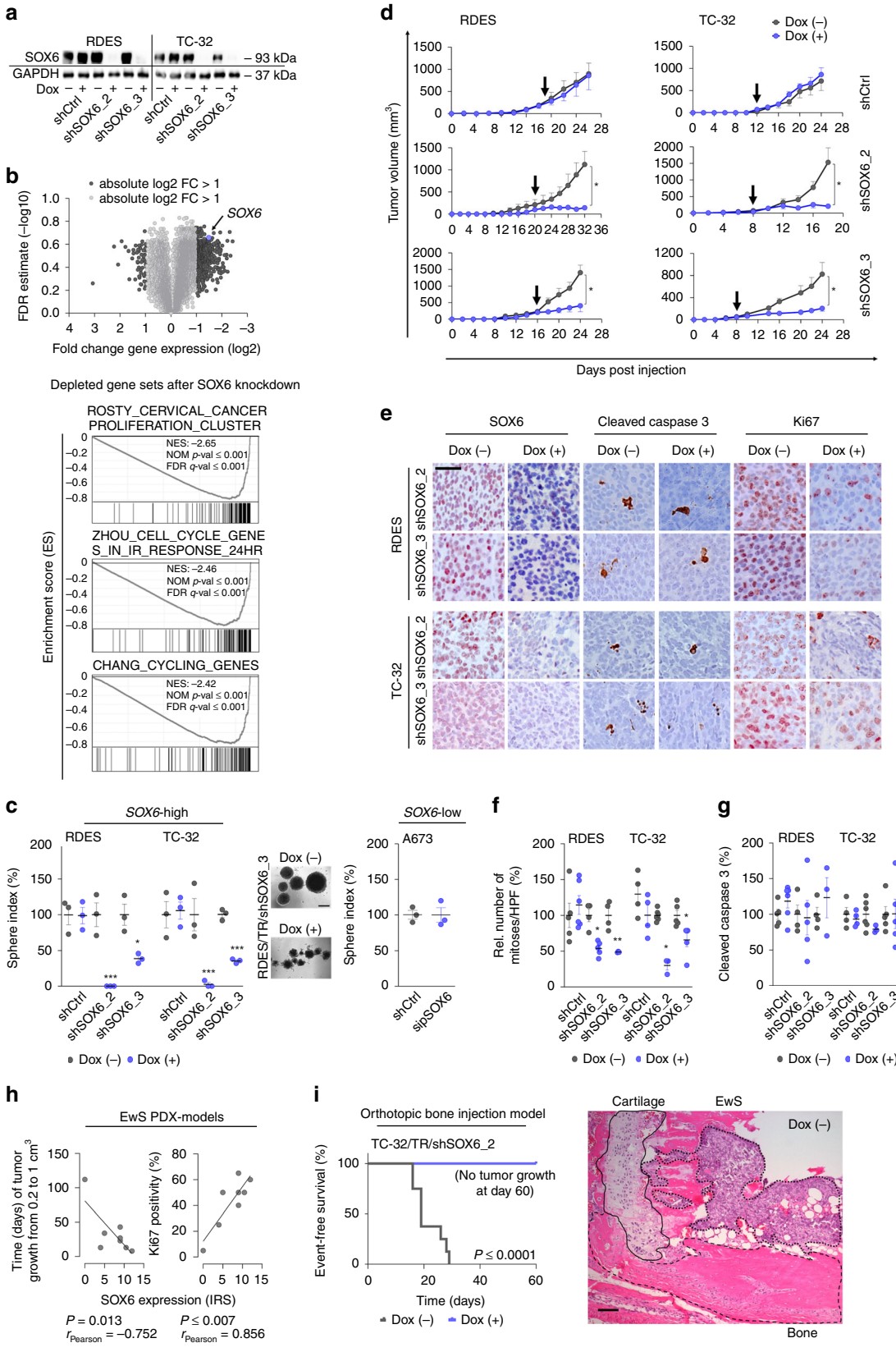

two different EwS cell lines with Dox-inducible shRNAs against *SOX6* or control shRNA subcutaneously into the flanks of NSG mice. While no effect of Dox-treatment was apparent in EwS cell lines expressing the non-targeting control shRNA, we noted a strong and consistent reduction of tumor growth upon *SOX6*

knockdown in both shRNA constructs and both cell lines (Fig. 3d). The knockdown of *SOX6* was confirmed ex vivo in xenografts by qRT-PCR (Supplementary Fig. 3f) and by immunohistochemistry (IHC) (Fig. 3e). Immunohistological assessment showed that *SOX6* silencing was associated with a

**Fig. 3 SOX6 promotes proliferation of EwS cells in vitro and in vivo. a** Representative SOX6 Western blot of $n = 3$ biologically independent replicates. **b** Top: Volcano plot of differentially expressed genes. Bottom: Representative GSEA plots. NES, normalized enrichment score; NOM, nominal $P$ value (determined by two-sided permutation testing without adjustment for multiple testing)[64]; FDR, false discovery rate. **c** Sphere index after SOX6 knockdown. Horizontal bars represent means, $n = 3$ biologically independent experiments. $P$ values determined via two-sided Mann–Whitney test. RDES: shSOX6_2 ($P \le 0.0001$), shSOX6_3 ($P = 0.018$); TC-32: shSOX6_2 ($P \le 0.0001$), shSOX6_3 ($P \le 0.0001$). Representative micrographs, scale bar = 1 mm. **d** Growth of EwS xenografts (arrow = start of Dox-treatment). Data are represented as means ($n$ indicates the number of biologically independent animals/group, TC-32: shCtrl ($n = 4$), shSOX6_2 (Dox(−) $n = 6$, Dox(+) $n = 3$, $P = 0.048$), shSOX6_3 ($n = 5$, $P = 0.032$); RDES: shCtrl (Dox(−) $n = 5$, Dox(+) $n = 6$), shSOX6_2 (Dox(−) $n = 4$, Dox(+) $n = 5$, $P = 0.019$), shSOX6_3 (Dox(−) $n = 4$, Dox(+) $n = 3$, $P = 0.029$)). $P$ values determined via two-sided Mann–Whitney test. **e** Representative IHC staining, scale bar = 20 μm. **f** Relative number of mitoses. Horizontal bars represent means ($n$ indicates the number of biologically independent animals/group, TC-32: shCtrl ($n = 4$), shSOX6_2 (Dox(−) $n = 6$, Dox(+) $n = 3$, $P = 0.019$), shSOX6_3 ($n = 5$, $P = 0.029$); RDES: shCtrl (Dox(−) $n = 5$, Dox(+) $n = 6$), shSOX6_2 (Dox(−) $n = 4$, Dox(+) $n = 5$, $P = 0.024$), shSOX6_3 (Dox(−) $n = 4$, Dox(+) $n = 3$, $P = 0.008$)). $P$ values determined via two-sided Mann–Whitney test. **g** Relative cleaved caspase 3 positivity. Horizontal bars represent means ($n$ indicates the number of biologically independent animals/group, TC-32: shCtrl ($n = 4$), shSOX6_2 (Dox(−) $n = 6$, Dox(+) $n = 3$), shSOX6_3 ($n = 5$); RDES: shCtrl (Dox(−) $n = 5$, Dox(+) $n = 6$), shSOX6_2 (Dox(−) $n = 4$, Dox(+) $n = 5$), shSOX6_3 (Dox(−) $n = 4$, Dox(+) $n = 3$)). **h** Correlation of time of growth of PDX models or Ki67 positivity with SOX6 expression ($n = 8$ biologically independent PDX). Lines indicate linear regressions. $P$ values determined by two-sided Pearson correlation test. **i** Event-free survival in orthotopic xenografts, $n = 8$ biologically independent animals/group. Representative micrograph of the injection site (Dox (−)), scale bar = 1000 μm. $P$ values determined via Mantel–Haenszel test ($P \le 0.0001$). All error bars represent SEM. ****$P \le 0.0001$, ***$P < 0.001$, **$P < 0.01$, *$P < 0.05$. Source data are provided as a Source Data file.

significant reduction of proliferation as indicated by Ki67 staining (Supplementary Fig. 3g) and numbers of mitotic cells per high-power field (HPF) (Fig. 3f), confirming the in vitro findings on SOX6 and cell cycle progression. In contrast, no significant differences in cleaved caspase 3 (Fig. 3g) and Annexin V staining (Supplementary Fig. 3h) were observed suggesting that the apparent reduction of tumor growth was not mediated by apoptotic cell death.

Notably, SOX6 protein expression was significantly positively correlated with the speed of tumor growth and Ki67 immunoreactivity in EwS patient-derived xenograft (PDX) models (Fig. 3h), in which we observed a broad range of SOX6 expression levels similar to that in primary EwS tumors (Fig. 1a, b). In line with this finding, SOX6 knockdown completely abolished the growth of TC-32/TR/shSOX6 EwS cells in an orthotopic tibial bone injection model in vivo (Fig. 3i).

Among the proliferation-associated genes downregulated after SOX6 knockdown (Fig. 3b), three genes (CDCA3, DEPDC1 and E2F8) appeared as plausible candidate genes to promote the pro-proliferative phenotype of SOX6, as they were previously shown to be involved in cell cycle progression[28–30].

In accordance, knockdown of any one of these genes with specific sipools in RDES and TC-32 EwS cells phenocopied, at least in part, the anti-proliferative effect of SOX6 knockdown (Supplementary Fig. 3i,j).

Collectively, these results highlight a contribution of SOX6 to proliferation, clonogenic growth and tumorigenicity of EwS cells in vitro and in vivo.

**SOX6 expression confers sensitivity toward Elesclomol.** To explore whether high SOX6 levels could constitute a specific vulnerability of EwS that may be exploited therapeutically, we interrogated a published gene expression dataset with matched drug-response data comprising up to 22 EwS cell lines[31]. To this end, we calculated for all 264 tested drugs the Pearson correlation coefficient and its statistical significance of the corresponding IC50 values with the observed SOX6 expression levels across EwS cell lines (Fig. 4a). Among the top seven drugs, Elesclomol (N-malonyl-bis (N-methyl-N-thiobenzoyl hydrazide) ($r_{Pearson} = -0.565$; $P = 0.014$) was the only drug, which effectively could inhibit EwS growth at a nanomolar range (median IC50 across cell lines ~25 nM) (Fig. 4b). Elesclomol is a potent oxidative stress inducer, which can bind copper in the Cu(II) state. The Elesclomol–copper complex then undergoes a redox reaction, and copper is reduced to the Cu(I) state. In turn, this

phenomenon produces free radicals by a Fenton reaction[32,33]. In cancer cells, the elevation of oxidative stress levels beyond a tolerable threshold exerts pro-apoptotic effects[34]. Indeed, in validation drug-response assays, Elesclomol strongly decreased viability of EwS cells with high SOX6 levels while the osteosarcoma cell line SAOS-2 and non-transformed human primary MSC line MSC-52 that exhibit low SOX6 expression levels were relatively resistant (Fig. 4c, d). Likewise, the SOX6-low expressing EwS cell line A673 exhibited a similarly low sensitivity toward Elesclomol than SAOS-2 and MSC-52 (Fig. 4c, d). The high sensitivity of SOX6-high expressing EwS cells toward Elesclomol appeared to be independent of proliferation under normal conditions, since the osteosarcoma cell line SAOS-2 and the SOX6-low expressing EwS cell line A673 proliferated even more than the tested SOX6-high expressing EwS cells (Fig. 4e). Yet, knockdown of SOX6 in RDES and TC-32 EwS cells significantly diminished their sensitivity toward Elesclomol (Fig. 4f), pointing to a functional role of SOX6 in Elesclomol sensitivity.

Consistent with a prior report in other cancer cell lines[34], Elesclomol strongly induced apoptosis in EwS cell lines in vitro when treated with corresponding IC50 concentrations (Fig. 4g), without affecting SOX6 expression levels (Supplementary Fig. 4a). In accordance, intravenous administration of Elesclomol for nine days reduced local tumor growth of TC-32 EwS xenografts in vivo (Fig. 4h), which was accompanied by induction of apoptosis and cell death as indicated by increased numbers of cells positive for cleaved caspase 3 (Fig. 4i), and more necrotic tumor area (Fig. 4j). Of note, mice treated with Elesclomol did not exhibit overt adverse effects such as weight-loss (Supplementary Fig. 4b). Similar to these in vivo models, Elesclomol-treatment strongly reduced growth of EwS cells compared to DMSO controls in a 3D in vitro mineralized bone model, and strongly induced apoptosis as evidenced by staining for cleaved caspase 3 (Fig. 4k).

In sum, these results demonstrate that SOX6 expression in EwS cells confers a proliferation-independent sensitivity toward the oxidative stress-inducing small-molecule Elesclomol.

**Elevated oxidative stress mediates Elesclomol sensitivity.** Since Elesclomol can induce oxidative stress, we investigated whether Elesclomol-treatment modulates oxidative stress in EwS cell lines and why EwS cells are sensitive to Elesclomol. Indeed, in both EwS cell lines, treatment of RDES and TC-32 cells with Elesclomol (10 nM) significantly induced oxidative stress compared to control (DMSO) as determined by analysis of DCF-DA to DFA conversion (Fig. 5a). Since Elesclomol may be able to interact

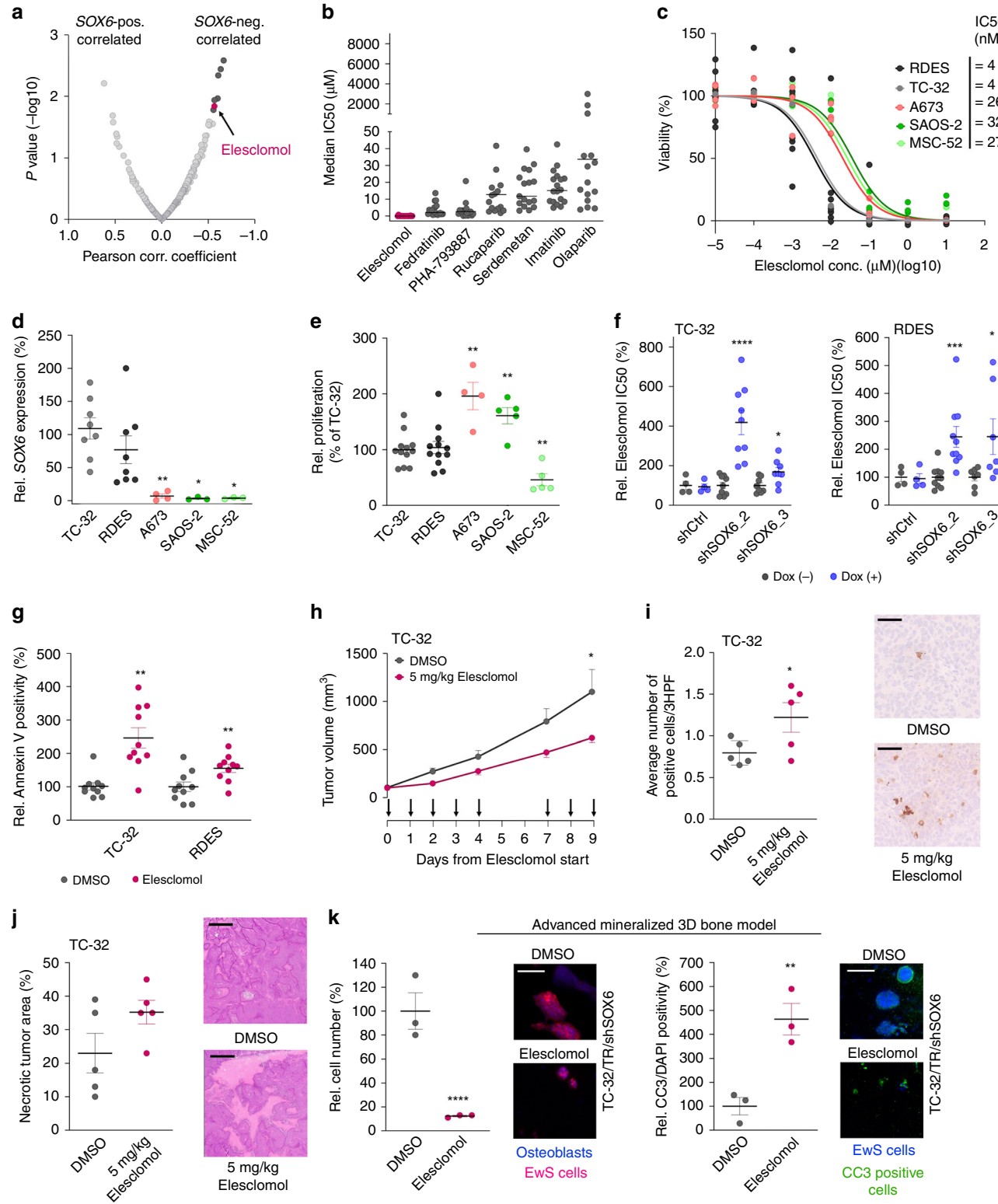

with the electron transport chain and to strongly induce intra-mitochondrial reactive oxygen species (mito-ROS)[32,33], we stained EwS cells after Elesclomol-treatment with MitoSOX Red, which can detect mitochondrial superoxide anions[35], and normalized the resulting MitoSOX Red fluorescence signal to the mitochondrial mass as determined by MitoTracker Green stains[35]. Indeed, we found that Elesclomol-treatment significantly induced the relative fluorescence signal for MitoSOX Red

(Fig. 5b), which may suggest that at least part of the oxidative stress induced by Elesclomol is derived from mito-ROS.

To test whether elevated oxidative stress levels play a role in the capacity of Elesclomol to kill EwS cells, we carried out drug-response assays in the presence/absence of the antioxidants N-acetyl cysteine (Nac)[36] and Tiron[37]. In both EwS cell lines, Nac- or Tiron-treatment resulted in significantly reduced conversion of DCF-DA to DCF (Fig. 5c) and decreased Elesclomol sensitivity

**Fig. 4 SOX6 expression confers sensitivity toward Elesclomol. a** Correlated expression/drug-response data[31]. Dark gray/pink = top-7 drugs (two-tailed Pearson correlation without adjustment for multiple testing). **b** IC50 of top-7 drugs (*n* indicates the number of independent cell lines: Elesclomol, *n* = 18, *P* = 0.015; Fedratinib, *n* = 19, *P* = 0.012; PHA-793887, *n* = 19, *P* = 0.004; Rucaparib, *n* = 18, *P* = 0.003; Serdemetan, *n* = 17, *P* = 0.011; Imatinib, *n* = 18, *P* = 0.017; Olaparib, *n* = 20, *P* = 0.005). Horizontal bars represent medians. **c** Relative viability (*n* indicates the number of biologically independent experiments/cell line: RDES (black) *n* = 10, TC-32 (gray) *n* = 9, A673 (orange) *n* = 4, SAOS-2 (green) *n* = 4, MSC-52 (light green) *n* = 3). **d** *SOX6* expression. Horizontal bars represent means (*n* indicates the number of biologically independent experiments/cell line: TC-32 *n* = 8, RDES *n* = 8, A673 *n* = 4 (*P* = 0.004), SAOS-2 *n* = 3 (*P* = 0.012), MSC-52 *n* = 3 (*P* = 0.012)). *P* values determined via two-sided Mann–Whitney test. **e** Cell viability. Horizontal bars represent means (*n* indicates the number of biologically independent experiments/cell line: TC-32 *n* = 12, RDES *n* = 12, A673 *n* = 4 (*P* = 0.004), SAOS-2 *n* = 5 (*P* = 0.006), MSC-52 *n* = 5 (*P* = 0.006)). *P* values determined via two-sided Mann–Whitney test. **f** Elesclomol-IC50. Horizontal bars represent means (*n* indicates the number of biologically independent experiments/cell line: TC-32: shCtrl *n* = 4, shSOX6_2 *n* = 9 (*P* ≤ 0.0001), shSOX6_3 *n* = 6 (*P* = 0.012); RDES: shCtrl *n* = 4, shSOX6_2 *n* = 10 (*P* = 0.0007), shSOX6_3 *n* = 5 (*P* = 0.026)). *P* values determined via two-sided Mann–Whitney test. **g** Annexin-V-positivity. Horizontal bars represent means, *n* = 10 biologically independent experiments. *P* values determined via unpaired two-sided *t*-test with Welch-correction; TC-32 (*P* = 0.0009), RDES (*P* = 0.009). **h** EwS-xenografts under Elesclomol-treatment (arrows). Data represent means, *n* = 5 biologically independent animals/condition. *P* value determined via two-sided Mann–Whitney test (*P* = 0.046). **i** Left: CC3-positivity. Horizontal bars represent means, *n* = 5 biologically independent xenografts/condition. *P* value determined via one-sided Mann–Whitney test (*P* = 0.044). Right: Representative micrographs, scale bar = 100 μm. **j** Left: Relative necrosis. Horizontal bars represent means, *n* = 5 biologically independent xenografts/condition. Right: Representative micrographs, scale bar = 900 μm. **k** Left: Relative cellularity. Horizontal bars represent means, *n* = 3 biologically independent experiments. Representative micrographs, scale bar = 50 μm. *P* value determined via two-sided Mann–Whitney test (*P* ≤ 0.0001). Right: CC3/DAPI-positivity. Horizontal bars represent means, *n* = 3 biologically independent experiments. Representative micrographs, scale bar = 50 μm. *P* value determined via two-sided Mann–Whitney test (*P* = 0.002). All error bars represent SEM. ****P* ≤ 0.0001, ***P* < 0.001, ***P* < 0.01, *P* < 0.05. Source data are provided as a Source Data file.

(Fig. 5d), indicating that this drug may exert its pro-apoptotic effect in EwS to a certain degree via induction of oxidative stress over a tolerable threshold.

In line with this hypothesis, Dox-induced knockdown of *SOX6* reduced the rate of DCF-DA to DFA conversion as well as the relative levels of MitoSOX Red (Fig. 5e, f) in both EwS cell lines (RDES and TC-32) and for both shRNA constructs, which was not observed in corresponding controls (shCtrl). To functionally validate the potential oxidative stress-dependent sensitivity toward Elesclomol conferred by SOX6 on EwS cells, we performed rescue experiments. In those, we noted that addition of the potent oxidative ROS-inducer $H_2O_2$ on the SOX6-silenced EwS cells could fully restore the sensitivity of these cell lines toward Elesclomol (Fig. 5g), while having no effect on viability of EwS cells that were not treated with Elesclomol (Supplementary Fig. 5a).

Together, these data suggested that SOX6 is involved in oxidative stress regulation in vitro and in vivo, and that the cellular oxidative stress levels may influence the sensitivity of EwS toward Elesclomol.

**SOX6-mediated TXNIP expression causes Elesclomol sensitivity.** To explore possible links between SOX6, oxidative stress, and Elesclomol sensitivity in EwS cells, we reassessed our micro-array data obtained from EwS cells with/without knockdown of SOX6. Of note, *TXNIP* (thioredoxin interacting protein)— a key inhibitor of the thioredoxin antioxidant system[38,39]—was the second most strongly downregulated gene after *SOX6* silencing in this discovery experiment (Supplementary Data 2).

We further investigated this finding by analyzing additional microarray gene expression data generated on the same microarray platform in 18 EwS cell lines (triplicate measurements per cell line). In this dataset, we found a significant positive correlation ($r_{Pearson}$ = 0.559, *P* = 0.016) of the average *SOX6* and *TXNIP* mRNA expression levels across EwS cell lines (Fig. 6a).

To corroborate the link between SOX6 and TXNIP, we carried out multiple independent SOX6 knockdown experiments in which we detected a consistent downregulation of TXNIP at the mRNA and protein level in vitro (Fig. 6b). In addition, we found a significant downregulation of TXNIP in vivo by immunohistochemistry in xenografts with SOX6 knockdown (Fig. 6c).

Interestingly, direct siRNA-mediated knockdown of *TXNIP* in EwS cells (Fig. 6d) significantly reduced the conversion rate of DCF-DA to DCF (Fig. 6e), and the relative levels of MitoSOX Red fluorescence (Fig. 6f). In accordance, direct knockdown of *TXNIP* reduced the sensitivity of EwS cells toward Elesclomol (Fig. 6g) to a similar extent than *SOX6* knockdown (Fig. 4f). Even more strikingly, inducible re-expression of *TXNIP* in SOX6-silenced EwS cells onto 'physiological' levels was sufficient to rescue ~66% of the sensitivity of these EwS cells toward Elesclomol (Fig. 6h). Of note, inducible overexpression of *SOX6* in the *SOX6*-low expressing EwS cell line A673 was accompanied by a significant induction of *TXNIP* expression and increased sensitivity toward Elesclomol (*P* < 0.01) (Supplementary Fig. 5b, c).

Taken together, these data provide evidence that the EWSR1-FLI1-induced strong upregulation of SOX6 contributes, at least in part, via upregulation of TXNIP and oxidative stress levels to the high sensitivity of EwS cells toward the oxidative stress-inducing drug Elesclomol (Fig. 6i).

## Discussion

EwS is a highly aggressive bone or soft-tissue cancer, possibly descending from osteo-chondrogenic progenitors. Since the transcription factor SOX6 is crucial for endochondral ossification and thus for bone development[16,40], we aimed at analyzing its role in EwS.

Our results show that *SOX6* is a direct EWSR1-FLI1 target gene that is highly but variably overexpressed at the mRNA and protein level in EwS as compared to most normal tissues and other cancers. However, we found no correlation of copy number variations and differences in promoter methylation with *SOX6* expression levels in EwS tumors. In contrast, we identified an intronic *SOX6*-associated GGAA-mSat that was bound by EWSR1-FLI1 in vivo and which exhibited strong length- and EWSR1-FLI1-dependent enhancer activity in EwS cell lines. Thus, it is tempting to speculate that the observed inter-tumor heterogeneity in SOX6 expression of EwS tumors and EwS cell lines is likely caused by inter-individual differences in the number of consecutive GGAA-repeats at this *SOX6*-associated GGAA-mSat. These findings are in line with recent observations for other direct EWSR1-FLI1 target genes such as *EGR2*, *MYBL2*, and *NR0B1* whose variable expression in EwS tumors is caused by inter-individual differences in

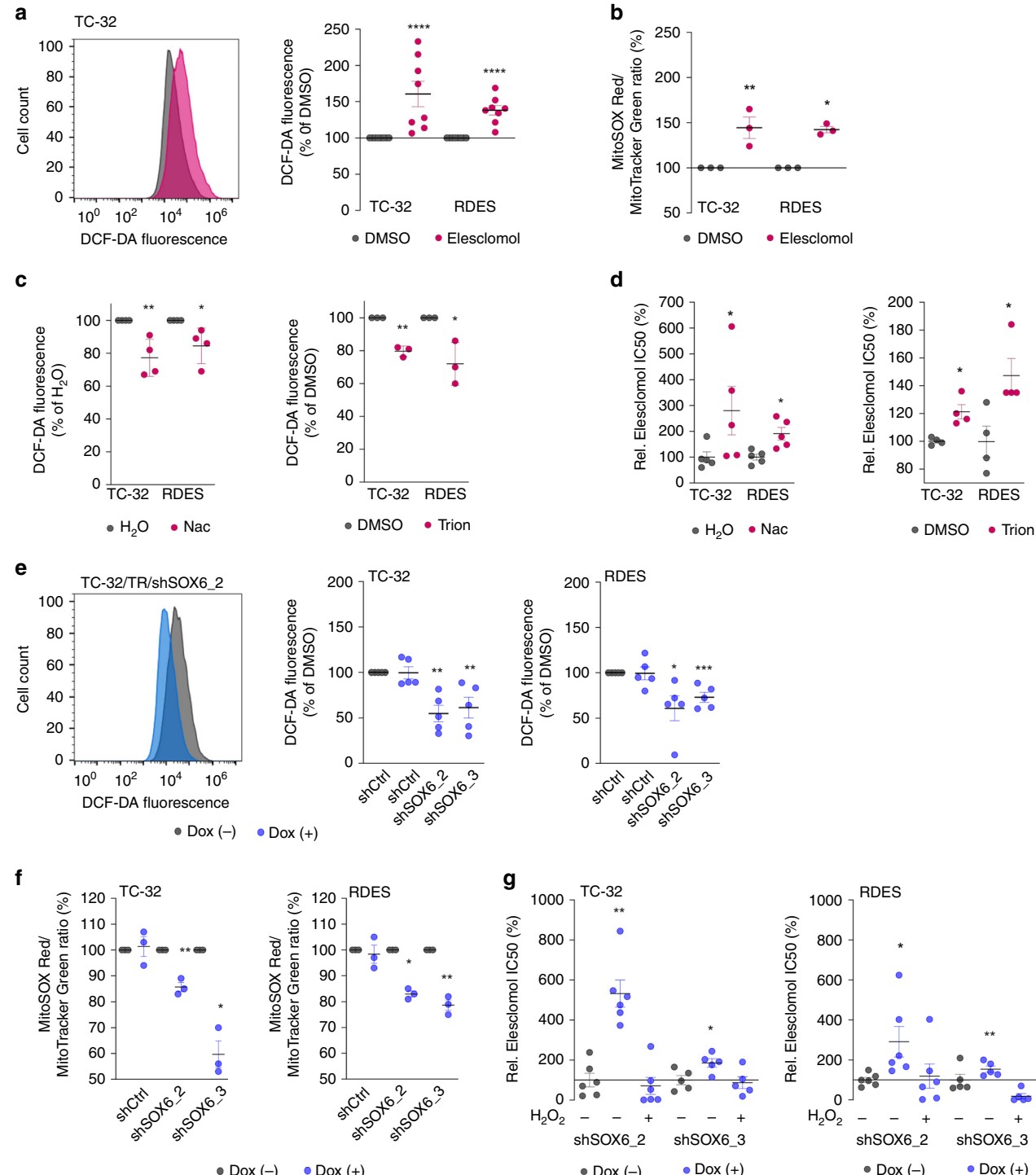

GGAA-repeat numbers of the corresponding enhancer-like GGAA-mSat[21,22,41].

Depending on the cellular context, SOX6 may act as a splicing factor[23,24] or as transcriptional regulator[40]. In transcriptome profiling experiments that comprised >285,000 transcripts and isoforms, we did not observe a strong contribution of SOX6 to alternative splicing in EwS. Instead, we identified a broad gene expression alteration after SOX6 silencing, pointing to a more pronounced role of SOX6 as a transcription factor in EwS. Especially the downregulated DEGs after *SOX6* knockdown were significantly enriched for gene sets involved in proliferation and cell cycle progression. These changes in the cellular transcriptome

were mirrored in functional in vitro and in vivo experiments. The strong pro-tumorigenic function of SOX6 in EwS is intriguing since in other cancer entities such as esophageal squamous cell carcinoma and hepatocellular carcinoma *SOX6* was reported to act as a tumor suppressor[42,43]. However, in EwS, knockdown of *SOX6* strongly reduced anchorage-independent growth and tumorigenicity, which was accompanied by delayed transition through cell cycle phases, reduced mitotic index and decreased expression of the proliferation marker Ki67. Thus, our results suggest that SOX6 may also have oncogenic properties, and that its oncogenic or tumor-suppressive function may depend on the cellular context. In this regard, it is interesting to note that other

**Fig. 5 Elesclomol sensitivity is induced by oxidative stress levels. a** Representative histogram (left) and quantification (right) of DCF-DA fluorescence in TC-32 cells after Elesclomol-treatment (10 nM, pink) compared to DMSO control (gray). Horizontal bars represent means, $n = 8$ biologically independent experiments. $P$ values determined via two-sided independent one-sample $t$-test (TC-32 ($P \leq 0.0001$), RDES ($P \leq 0.0001$). **b** MitoSOX Red /MitoTracker Green ratio in EwS cells after Elesclomol-treatment (10 nM). Horizontal bars represent means, $n = 3$ biologically independent experiments. $P$ values determined via two-sided independent one-sample $t$-test (TC-32 ($P = 0.006$), RDES ($P = 0.025$). **c** Relative DCF-DA fluorescence in EwS cells after antioxidant-treatment (0.01 mM Nac or 0.1 mM Tiron). Horizontal bars represent means ($n$ indicates the number of biologically independent experiments per condition: Nac: TC-32 ($n = 4$, $P = 0.008$), RDES ($n = 4$, $P = 0.018$), Tiron: TC-32 ($n = 3$, $P = 0.005$), RDES ($n = 3$, $P = 0.045$)). $P$ values determined via two-sided independent one-sample $t$-test. **d** Elesclomol IC50 in EwS cells after pre-treatment with Nac (0.01 mM) or Tiron (0.1 mM). Horizontal bars represent means ($n$ indicates the number of biologically independent experiments per condition: Nac: TC-32 ($n = 5$, $P = 0.032$), RDES ($n = 5$, $P = 0.016$); Tiron: TC-32 ($n = 4$, $P = 0.029$), RDES ($n = 4$, $P = 0.029$)). $P$ values determined via two-sided Mann–Whitney test. **e** Representative histogram of DCF-DA fluorescence (left) and quantification (right) in TC-32/TR/shSOX6_2 cells 96 h after SOX6 knockdown. Horizontal bars represent means, $n = 5$ biologically independent experiments. $P$ values determined via two-sided independent one-sample $t$-test (TC-32: shSOX6_2 ($P = 0.001$), shSOX6_3 ($P = 0.005$); RDES: shSOX6_2 ($P = 0.025$), shSOX6_3 ($P = 0.001$)). **f** Relative MitoSOX Red/MitoTracker Green ratios in EwS cells 96 h after SOX6 knockdown. Horizontal bars represent means, $n = 3$ biologically independent experiments. $P$ values determined via two-sided independent one-sample $t$-test (TC-32: shSOX6_2 ($P = 0.008$), shSOX6_3 ($P = 0.014$); RDES: shSOX6_2 ($P = 0.038$), shSOX6_3 ($P = 0.006$)). **g** Relative Elesclomol IC50 in EwS cells after SOX6 knockdown and treatment with $H_2O_2$ (30 μmol/l) for 72 h. Horizontal bars represent means ($n$ indicates the number of biologically independent experiments per condition: TC-32: shSOX6_2 ($n = 6$, $P = 0.002$) shSOX6_3 ($n = 5$, $P = 0.032$); RDES: shSOX6_2 ($n = 6$, $P = 0.041$), shSOX6_3 ($n = 5$; $P = 0.008$)). $P$ values determined via two-sided Mann–Whitney test. All error bars represent SEM. ****$P \leq 0.0001$, ***$P < 0.001$, **$P < 0.01$, *$P < 0.05$. Source data are provided as a Source Data file.

direct EWSR1-FLI1 target genes, such as *EGR2*, *MYBL2*, and *NR0B1*, which are driven by GGAA-mSats, are also critical mediators of EwS tumorigenesis[21,22,44].

Since therapeutic options for EwS patients are urgently required[1], we investigated whether the high expression of SOX6 in EwS may provide a vulnerability that could be exploited therapeutically. Indeed, we discovered that high expression of SOX6 confers hypersensitivity toward the small-molecule Elesclomol. While Elesclomol was shown to inhibit cancer cell growth in vitro at micro-molar concentrations, in melanoma, breast cancer and leukemia cell lines[32,33,45,46] we noted a much higher sensitivity of EwS cells toward Elesclomol with IC50 values in the nanomolar range. These observations suggest that the higher sensitivity of EwS toward Elesclomol may be caused by the relatively higher expression of *SOX6* in EwS compared to other cancers such as osteosarcoma and melanoma (Supplementary Fig. 4c). In support of this hypothesis, Elesclomol-treatment in combination with paclitaxel had only moderate effect on outcome of unselected patients affected by malignant melanoma in phase II and III clinical trials[47,48], and may have shown higher efficacy when preselecting patients with higher SOX6 levels or higher intracellular oxidative stress levels.

Since clinical development of Elesclomol has been discontinued, we explored whether other compounds could serve as alternatives. Interestingly, among three so-called oxidative stress inducers (BRD56491, DC_AC50, and Menadione), BRD56491 and DC_AC50 failed to induce oxidative stress and subsequent cell death at physiologically relevant concentrations in EwS cells (IC50 > 20 μM) (Supplementary Fig. 6a). In contrast, Menadione was able to induce oxidative stress (including mito-ROS) (Supplementary Fig. 6b) and to decrease the viability of EwS cells expressing high *SOX6* levels, albeit only at ~1500-fold higher concentrations compared to Elesclomol (IC50 ~6 μM for Menadione vs. ~4 nM for Elesclomol) (Supplementary Fig. 6c), which may highlight the potency of Elesclomol.

Since many cancer types including EwS display an oxidative stress phenotype characterized by higher oxidative stress levels than normal tissues[49,50], cancer cells tend to be more sensitive toward further increases in oxidative stress than nonmalignant cells[51]. Previous reports demonstrated that Elesclomol can induce oxidative stress beyond a tolerable threshold triggering apoptosis[32,34,52]. In line with these findings, we observed an increase of DCF-DA to DCF conversion and relative MitoSOX Red fluorescence followed by apoptosis after Elesclomol-treatment

in vitro and in vivo in EwS cells. The apoptotic effect appeared to be dependent on the baseline intracellular oxidative stress levels because knockdown of *SOX6* and concomitant downregulation of oxidative stress (as measured by DCF-DA to DCF conversion and relative MitoSOX Red fluorescence) or pre-treatment of EwS cells with the antioxidants Nac and Tiron diminished the sensitivity of EwS cells toward Elesclomol, whereas addition of the ROS-inducer $H_2O_2$ rescued the *SOX6* knockdown effect on Elesclomol sensitivity.

The elevated intracellular oxidative stress and mito-ROS levels as well as the associated hypersensitivity of *SOX6*-high expressing EwS cells toward Elesclomol can be explained, at least in part, by the SOX6-mediated upregulation of TXNIP. In fact, *TXNIP* knockdown in EwS cells was associated with decreased conversion rates of DCF-DA to DCF and lower levels of relative MitoSOX Red fluorescence, which might be caused by its potential inhibitory function on the thioredoxin (TRX) antioxidant system that plays an essential role in buffering intracellular oxidative stress[53,54]. In our rescue experiments, we noted that restoration of *TXNIP* expression (Fig. 6h) in SOX6-silenced EwS cells could rescue ~66% of the Elesclomol sensitivity. These data indicate that TXNIP is an important downstream mediator of the SOX6-induced Elesclomol sensitivity of EwS cells, but also suggest that other factors may play a role.

Besides, the role of TXNIP in EwS may extend beyond the TRX antioxidant system, since TXNIP has also been reported to impact on glucose and lipid metabolism[38]. While the potentially pro-proliferative role of TXNIP in EwS remains to be fully elucidated, it should be noted that in other cancers including hepatocellular, breast, and bladder carcinoma as well as leukemia, TXNIP has been reported to act as a tumor suppressor[39], indicating that this protein may, like SOX6, either promote or inhibit tumor growth depending on the cellular context.

Our data suggest that oxidative stress-inducing drugs such as Elesclomol could offer a therapeutic option for EwS patients with high SOX6 expression levels. Additionally, SOX6 may serve as a biomarker to predict the efficacy of Elesclomol-treatment in EwS, and perhaps other cancer types. Interestingly, Elesclomol-treatment has been shown to potentiate the pro-apoptotic effect of oxidative stress-dependent chemotherapeutic drugs such as doxorubicin in breast cancer[46]. Since doxorubicin is part of current standard treatment regimens for EwS patients, it is tempting to speculate that Elesclomol-treatment may also serve as an enhancer for doxorubicin treatment in EwS, even in patients

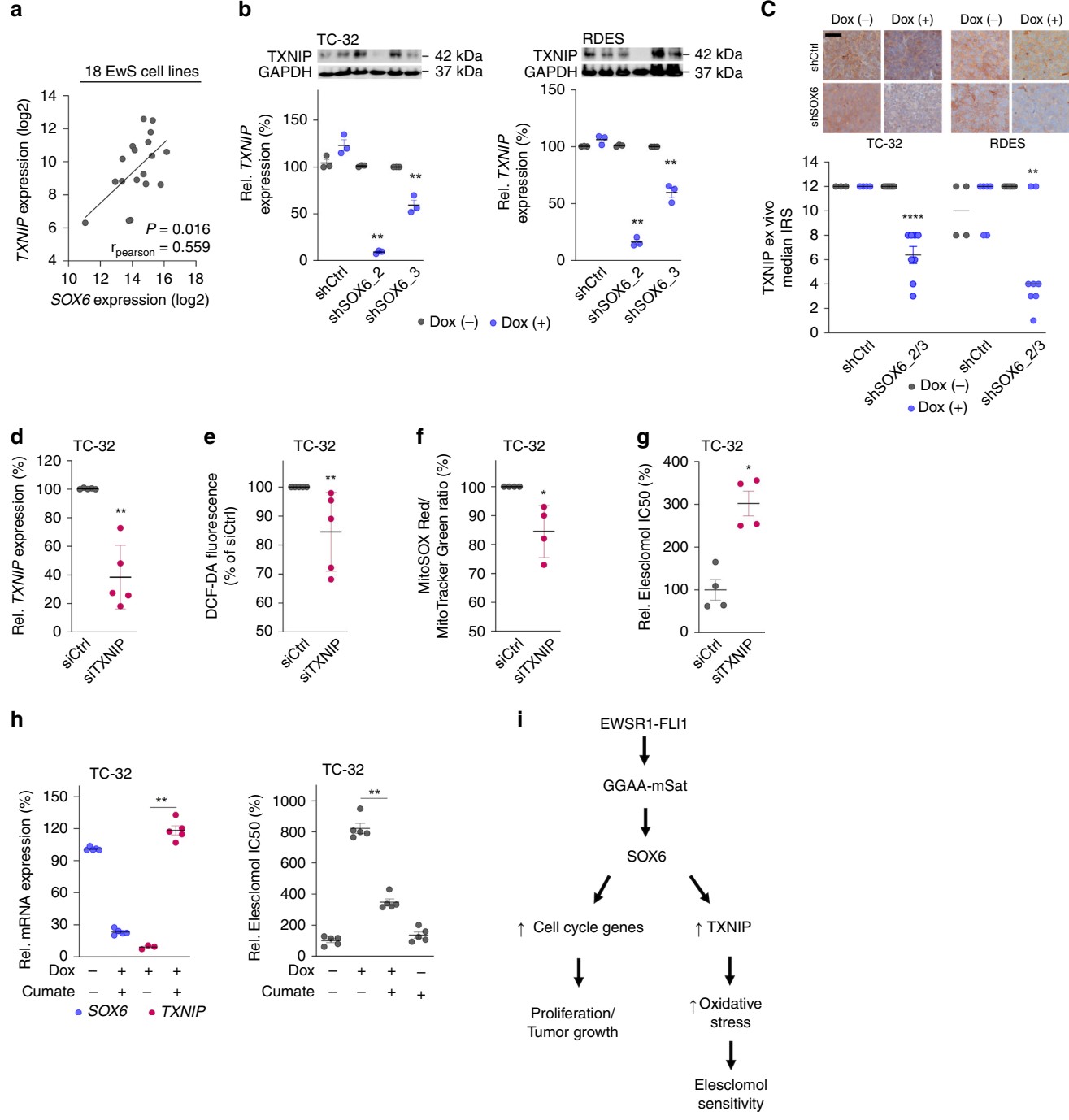

with relatively low intratumoral SOX6 levels. However, as Elesclomol targets actively respiring mitochondria, it may have less efficacy in hypoxic conditions[33], which may be the case in large hypoxic tumors.

In synopsis, we discovered that EWSR1-FLI1 hijacks SOX6 in EwS, which promotes tumor growth and modulates intracellular oxidative stress levels creating a therapeutic vulnerability toward oxidative stress-inducing drugs. Our results exemplify how aberrant activation of a developmental transcription factor by a dominant oncogene can promote malignancy but provide opportunities for targeted therapy.

## Methods

**Cell lines and cell culture conditions.** The neuroblastoma cell line SK-N-AS as well as HEK293T were purchased from ATCC. Human EwS cell lines and other cell lines were provided by the following repositories and/or sources: A673 cells were purchased from ATCC. MHH-ES1, RDES, RH-1, SK-ES1, and SK-N-MC and cells were provided by the German collection of Microorganism and Cell Cultures (DSMZ). CHLA-10, CHLA-25, CHLA-32, CHLA-57, CHLA-99, COG-E-352, TC-32, TC-71, and TC-106 were kindly provided by the Children's Oncology Group (COG), and ES7, EW1, EW3, EW7, EW16, EW17, EW18, EW22, EW24, MIC, ORS, POE, SK-PN-DW, SK-PN-LI, and STA-ET1 as well as rhabdomyosarcoma (Rh4, Rh36) and neuroblastoma (TGW) cell lines cells were provided by O. Delattre (Institute Curie, Paris). A673/TR/shEF1 cells were provided by J. Alonso (Madrid, Spain)[55]. Human osteosarcoma cell lines SAOS-2 and U2OS were provided by DMSZ, and the MSC cell line MSC-52 was generated from bone marrow of an EwS patient (provided by U. Dirksen; Essen, Germany). All cell lines were cultured in RPMI-1640 medium with stable glutamine (Biochrom, Germany) supplemented with 10% tetracycline-free fetal bovine serum (Sigma-Aldrich, Germany), 100 U/ml penicillin and 100 µg/ml streptomycin (Merck, Germany) at 37 °C with 5% $CO_2$ in a humidified atmosphere. Cell lines were routinely tested for mycoplasma contamination by nested PCR, and cell line identity was regularly verified by STR-profiling.

**Fig. 6 SOX6-mediated TXNIP expression causes Elesclomol sensitivity. a** Correlation of *TXNIP* and *SOX6* expression in 18 EwS cell lines. The line indicates the linear regression of the data (two-tailed Pearson correlation test, $P = 0.016$, $r_{pearson} = 0.559$). **b** Top: Representative Western blot analysis of TXNIP 96 h after *SOX6* knockdown. GAPDH served as loading control. Bottom: Relative *TXNIP* expression (qRT-PCR). Horizontal bars represent means, $n = 3$ biologically independent experiments. *P* values determined via two-sided Mann–Whitney test (all $P = 0.002$). **c** Top: representative micrographs from EwS xenografts with/without SOX6 knockdown stained for TXNIP. Bottom: IRS for TXNIP (*n* indicates the number of biologically independent xenografts per condition: TC-32: shCtrl (Dox(−) $n = 3$, Dox(+) $n = 4$), shSOX6_2/3 (Dox(−) $n = 11$, Dox(+) $n = 8$, $P \leq 0.0001$); RDES: shCtrl (Dox(−) $n = 4$, Dox(+) $n = 6$), shSOX6_2/3 (Dox(−) $n = 7$, Dox(+) $n = 8$, $P = 0.007$). Horizontal bars represent medians. *P* values determined via two-sided Mann–Whitney test. Scale bar = 20 µm. **d** Relative *TXNIP* expression (qRT-PCR) in TC-32 cells 96 h after *TXNIP* knockdown. Horizontal bars represent means, $n = 5$ biologically independent experiments, $P = 0.008$. *P* value determined via two-sided Mann–Whitney test. **e** Relative DCF-DA fluorescence in TC-32 cells after *TXNIP* knockdown. Horizontal bars represent means, $n = 5$ biologically independent experiments, $P = 0.009$. *P* value determined via two-sided independent one-sample *t*-test. **f** MitoSOX Red/MitoTracker Green ratio in TC-32 cells after *TXNIP* knockdown. Horizontal bars represent means, $n = 4$ biologically independent experiments, $P = 0.012$. *P* value determined via two-sided independent one-sample *t*-test. **g** Elesclomol IC50 upon *TXNIP* knockdown in TC-32 cells. Horizontal bars represent means, $n = 4$ biologically independent experiments, $P = 0.029$. *P* value determined via two-sided Mann–Whitney test. **h** Left: Relative mRNA expression in TC-32 EwS cells after 96 h of concomitant Dox-treatment (0.1 µg/ml) and Cumate-treatment (5 µg/ml). Horizontal bars represent means, $n = 5$ biologically independent experiments, $P = 0.008$. Right: Relative Elesclomol IC50 in TC-32 cells 72 h after Elesclomol-treatment and concomitant addition of Dox (0.1 µg/ml) and Cumate (5 µg/ml). Horizontal bars represent means, $n = 5$ biologically independent experiments, $P = 0.008$. *P* values determined via two-sided Mann–Whitney test. **i** Schematic illustration of the EWSR1-FLI1-mediated effect of SOX6 expression. All error bars represent SEM. ****$P \leq 0.0001$, ***$P < 0.001$, **$P < 0.01$, *$P < 0.05$. Source data are provided as a Source Data file.

**RNA extraction, reverse transcription, and qRT-PCR**. Total RNA was isolated using the NucleoSpin RNA kit (Macherey-Nagel, Germany). 1 µg of total RNA was reverse-transcribed using High-Capacity cDNA Reverse Transcription Kit (Applied Biosystems, USA). qRT-PCR reactions were performed using SYBR green Mastermix (Applied Biosystems) mixed with diluted cDNA (1:10) and 0.5 µM forward and reverse primer (total reaction volume 15 µl) on a BioRad CFX Connect instrument and analyzed using BioRad CFX Manager 3.1 software. Gene expression values were calculated using the $2^{-(\Delta\Delta Ct)}$ method[56] relative to the housekeeping gene *RPLP0* as internal control. Oligonucleotides were purchased from MWG Eurofins Genomics (Germany) and are listed in Supplementary Table 2. The thermal conditions for qRT-PCR were as follows: heat activation at 95 °C for 2 min, DNA denaturation at 95 °C for 10 s, and annealing and elongation at 60 °C for 20 s (50 cycles), final denaturation at 95 °C for 30 s.

**Transient RNA interference (RNAi)**. POE, RDES, TC-32, A673, and EW7 cells were transiently reversely transfected with Lipofectamine RNAiMAX according to the manufacturer's protocol (Invitrogen, USA) with either a non-targeting control sipool (sipCtrl) or sipools specifically directed against *CDCA3, DEPDC1, E2F8*, or *SOX6* (all siTOOLs, Biotech, Germany) or short interfering RNA (siRNA) against *TXNIP* (MWG Eurofins, Germany) (Supplementary Table 2) at a final concentration of 5–15 nM depending on the cell line and target gene. Cells were retransfected 48 h after the first transfection. All sipools consisted of 30 different siRNAs directed against the target transcript, which eliminates off-target effects[25]. Knockdown efficacy was validated by qRT-PCR and/or Western blot.

**Cloning of GGAA-mSats and luciferase reporter assays**. For luciferase reporter assays, both alleles of a 1-kb fragment including the *SOX6*-associated GGAA-mSat (hg19 coordinates: chr11: 16,394,850–16,395,749) were cloned from three highly *SOX6* expressing EwS cell lines (RDES, TC-32, POE) and three *SOX6* moderately (SK-N-MC, EW17, and ORS) and two *SOX6* lowly expressing EwS cells (A673, EW7) using GoTaq DNA Polymerase (Promega, Germany) and primer sequences listed in Supplementary Table 2. The thermal conditions for touchdown (TD)-PCR protocol were as follows: Initial denaturation: 95 °C for 2 min; denaturation 98 °C for 10 s; annealing: 59–49 °C for 30 s ($T_m$ (=59 °C) −5 °C + 10 °C higher for TD-PCR); − 1 °C every 2 cycles); extension: 72 °C for 1 min (2 × 10 cycles, and subsequently another 20 cycles at an annealing temperature of 65 °C); final extension: 72 °C for 5 min. These fragments were cloned upstream of the SV40 minimal promoter into the pGL3 luciferase reporter vector (Promega, #E1761, Germany) by In-Fusion HD Cloning Kit (Clontech, Japan) according to the manufacturer's protocol. These vectors were used for transformation of Stellar Competent Cells (Clontech); successfully transformed bacteria were selected using ampicillin (Merck). Correct insertion of the vector was confirmed by colony PCR. The number of consecutive GGAA-repeats for each allele and each cell line was determined by commercial Sanger sequencing. The sequencing primer is listed in Supplementary Table 2. The presence of additional genetic variation in the cloned sequences apart from the *SOX6*-associated GGAA-mSat was ruled out by whole-genome sequencing of the parental cell lines and confirmatory Sanger sequencing of the cloned fragments. For reporter assays, $5 × 10^4$ A673/TR/shEF1 cells per 24-well were seeded in 600 µl medium and transfected with pGL3 luciferase reporter vector and Renilla pGL3-Rluc (ratio, 100:1) using Lipofectamine LTX Reagent with PLUS Reagent (Invitrogen). Four hours after transfection, the culture media was replaced by media with/without (w/o) Doxycycline (Dox; 1 µg/ml; Merck). Cells were lysed after 72 h and monitored with a dual luciferase assay system (Berthold, Germany). Firefly luciferase activity was normalized to Renilla luciferase activity.

**Analysis of copy number variation and promoter methylation**. For analysis of genomic copy numbers, publicly available DNA copy number data for EwS tumors[57] with corresponding RNA expression data (GSE34620 and GSE37371, $n = 32$), were downloaded from the "soft-tissue cancer—Ewing sarcoma—FR" project from the International Cancer Genome Consortium (ICGC) Data Portal and Gene Expression Omnibus (GEO) of the NCBI, respectively. For the *SOX6* locus, segment mean values were extracted from these data using Visual Basic for Applications (VBA). The segment mean values were correlated with the log2-transformed expression of the candidate gene. For CpG methylation analysis, publicly available data on CpG methylation in 40 EwS tumors (GSE88826)[58] with corresponding RNA expression data (GSE34620) were downloaded from GEO. For the *SOX6* locus, the ratio of methylated versus unmethylated reads was calculated for two CpG sites (CpG1 hg19: chr11:15994482; CpG2 hg19: chr11:15994519) in each sample ($n = 40$) using VBA, which were covered by at least four reads.

**Analysis of *SOX6* expression levels in human embryoid bodies**. Publicly available gene expression microarray data for ectopic EWSR1-FLI1 expression in human embryoid bodies generated on the Affymetrix HG-U133Plus2.0 array (GSE64686)[59] were normalized by Robust Multiarray Average (RMA)[60] using custom brainarray chip description files (CDF; ENTREZG, v19) yielding one optimized probe-set per gene[61].

**Analysis of published DNase-Seq and ChIP-Seq data**. ENCODE SK-N-MC DNase-Seq (GSM736570) and ChIP-Seq data (GSE61944) were downloaded from the GEO, processed and displayed in the UCSC genome browser. The following samples were used in this study:
ENCODE_SKNMC_hg19_DNAseHS_rep1
GSM1517546_SKNMC.shGFP96.FLI1
GSM1517555_SKNMC.shFLI196.FLI1
GSM1517547_SKNMC.shGFP96.H3K27ac
GSM1517556_SKNMC.shFLI196.H3K27ac
GSM1517569_A673.shGFP48.FLI1
GSM1517572_A673.shFLI148.FLI1
GSM1517570_A673.shGFP48.H3K27ac
GSM1517573_A673.shFLI148.H3K27ac

**Transcriptome and splicing analyses**. To assess an impact of SOX6 on gene expression and on alternative splicing in EwS, microarray analysis was performed. To this end, $1.2 × 10^4$ cells were seeded in wells of 6-well plates and treated with Dox for 96 h (Dox-refreshment after 48 h). Thereafter, total RNA was extracted with the ReliaPrep miRNA Cell and Tissue Miniprep System (Promega) and transcriptome profiled at IMGM laboratories (Martinsried, Germany). RNA quality was assessed with a Bioanalyzer and samples with RNA integrity numbers (RIN) > 9 were hybridized to Human Affymetrix Clariom D microarrays. Data were quantile normalized with Transcriptome Analysis Console (v4.0; Thermo Fisher Scientific) using the SST-RMA algorithm. Annotation of the data was performed using the Affymetrix library for Clariom D Array (version 2, human), both on gene and exon level. DEGs with consistent and significant fold changes (FCs) across shRNAs and cell lines were identified as follows: First, normalized gene expression signal was log2 transformed. To avoid false discovery artifacts due to the detection of only minimally expressed genes, we excluded all genes with a lower or just minimally higher gene expression signal than that observed for *ERG* (mean log2 expression signal of 6.05), which is known to be virtually not expressed in EWSR1-FLI1 positive EwS cell lines[62]. Accordingly, only genes with mean log2

expression signals (w/o Dox condition) of at least 7 were further analyzed. The FCs of the shControl samples and both specific shRNAs were calculated for each cell line separately. Then the FCs observed in the respective shControl samples were subtracted from those seen in the shSOX6 samples, which yielded the FCs for each specific shRNAs in each cell line. Then these FCs were averaged to obtain the mean FC per gene across shRNAs and cell lines. As the log2 FC for *SOX6* was −1.486, we assumed that downstream targets should also be strongly regulated. Consequently, only genes with a minimum absolute log2 FC of 1 were considered as DEGs. Additionally, the false discovery rate (FDR) was estimated for each gene using the R package qvalue from BioConductor[63]. To identify enriched gene sets, genes were ranked by their expression FC between the groups Dox (−/+), and a pre-ranked GSEA (MSigDB v5.2, c2.cgp.all) with 1,000 permutations was performed[64].

To assess the potential role of SOX6 in alternative splicing, we assumed that, if RNA is not alternatively spliced, the ratio of each probe selection region (PSR) expression measured on the Affymetrix microarray with and without *SOX6* knockdown stays the same independently of up- or downregulation of the corresponding gene. Hence, we calculated the additional FC between the expression value of each PSR before and after SOX6 knockdown to the FC expected by expression regulation assessed on the gene level. Of 539,385 PSRs with 47,851 matched genes in our analysis, 22,155 PSRs (10,754 genes) showed a consistent positive or negative additional log2 transformed expression FC of ≥0.3. For 20,050 PSRs (10,179 genes) expression differences were significant ($P \leq 0.05$) when corrected for the FC on gene level. However, none of the PSRs remained significant after Bonferroni correction for multiple testing. Gene expression data were deposited at the GEO (accession code GSE120576).

For comparative analyses of *SOX6* and *TXNIP* mRNA expression levels, total RNA of 18 EwS cell lines (A673, CHLA-10, CHLA-25, EW1, EW22, EW24, EW3, EW7, MHH-ES1, MIC, POE, RDES, RH-1, SK-ES1, SK-N-MC, TC-106, TC-32, and TC-71) was assessed by Affymetrix Clariom D microarrays as described above in triplicate measurements per cell line.

**Generation of Dox-inducible shRNA-expressing cells.** For long-term experiments human EwS cell lines SK-N-MC, POE, RDES and TC-32 were transduced with lentiviral pLKO-TET-ON all-in-one vector system (plasmid #21915, Addgene) containing a puromycin-resistance cassette, and a tet-responsive element for Dox-inducible expression of shRNA against *EWSR1-FLI1* (shEF1), *SOX6* (shSOX6) or a non-targeting control shRNA (shCtrl). Dox-inducible vectors were generated according to a publicly available protocol[65] using In-Fusion HD Cloning Kit (Clontech) (Supplementary Table 2). Vectors were amplified in Stellar Competent Cells (Clontech) and integrated shRNA was verified by Sanger sequencing. The sequencing primer is listed in Supplementary Table 2. Lentiviral particles were generated in HEK293T cells. Virus-containing supernatant was collected to infect the human EwS cell lines. Successfully transduced cells were selected with 1.5 µg/ml puromycin (InVivoGen, USA). The shRNA expression for SOX6 knockdown in EwS cells was achieved by adding 0.1 µg/ml Dox every 48 h to the medium. Generated cell lines were designated as SK-N-MC/TR/shEF1, POE/TR/shSOX6_1, POE/TR/shSOX6_3, RDES/TR/shCtrl, RDES/TR/shSOX6_2, RDES/TR/shSOX6_3, TC-32/TR/shCtrl, TC-32/TR/shSOX6_2, and TC-32/TR/shSOX6_3.

**TXNIP rescue experiments.** For rescue experiments, TC-32/TR/shSOX6_2 EwS cells were additionally transduced with a Cumate-inducible overexpression system based on the pCDH-CuO-MCS-EF1α-CymR-T2A-Puro SparQ™ All-in-one Cloning and Expression Lentivector (System Biosciences, USA, QM800A-1-SBI)[66]. The full-length cDNA of *TXNIP* (NM_006472) was PCR-amplified from a commercial plasmid (GenScript, Hong Kong, OHu20973) and cloned into the multiple cloning site. In addition, the puromycin-resistance gene was replaced by an eGFP coding sequence cloned from pCMV-GFP plasmid (Addgene Plasmid #11153). The correct insertion of the full-length *TXNIP* cDNA was verified by Sanger sequencing. The PCR and sequencing primers are listed in Supplementary Table 2. Successfully transduced EwS cells were subsequently sorted by a fluorescence-activated cell sorter (FACS). Re-expression of *TXNIP* in these EwS cells was achieved by addition of 5 µg/ml Cumate to the culture medium (media change including Cumate every 48 h).

**Inducible overexpression of SOX6.** *SOX6* overexpression experiments were carried out in the *SOX6*-low expressing EwS cell line A673. To this end, full-length cDNA of *SOX6* (NM_033326.3) was PCR-amplified from a commercial plasmid (GenScript, OHu22944) and cloned into the multiple cloning site of a modified pTRIPZ vector[67]. The correct insertion was verified by Sanger sequencing. The PCR and sequencing primers are listed in Supplementary Table 2. A673 cells were transduced with this lentiviral vector containing a puromycin-resistance cassette, and a tet-responsive element for Dox-inducible expression of full-length cDNA of *SOX6* (A673/TR/cSOX6). Successfully transduced cells were selected with 0.5 µg/ml puromycin. Overexpression of *SOX6* was achieved by addition of 1 µg/ml Dox to the culture medium.

**Western blot.** RDES/TR/shCtrl, RDES/TR/shSOX6_2, RDES/TR/shSOX6_3, TC-32/TR/shCtrl, TC-32/TR/shSOX6_2, and TC-32/TR/shSOX6_3 EwS cells were treated for 96 h with Dox to induce SOX6 knockdown. Whole cellular protein was extracted

with RIPA buffer containing 1 mM $Na_3VO_4$ (Sigma-Aldrich) and protease inhibitor cocktail (Sigma-Aldrich). Western blots were incubated with mouse monoclonal anti-SOX6 antibody (1:1,000, sc-393314, Santa Cruz, Germany), rabbit monoclonal anti-TXNIP antibody (1:1,000, ab188865, Abcam, UK) and mouse monoclonal anti-GAPDH (1:800, sc-32233, Santa Cruz). The nitrocellulose membranes (GE Healthcare Biosciences, Germany) were secondary incubated with anti-mouse IgG (H + L) horseradish peroxidase coupled (1:3,000, W402b, Promega) and polyclonal anti-rabbit IgG (1:5000, R1364HRP, OriGene, Germany). Proteins were detected using chemiluminescence HRP substrate (Merck). Densitometric protein quantifications were carried out by ImageJ.

**Proliferation assays.** For proliferation assays, $2 \times 10^5$ EwS cells were seeded in wells of 6-well plates and treated with 0.1 µg/ml Dox every 48 h for knockdown or transiently transfected with Lipofectamine RNAiMAX (Invitrogen, USA) and the respective sipool every 48 h for a total period of 96 h. Cell viability was determined including the supernatant by counting the cells with Trypan-Blue (Sigma-Aldrich) in standardized hemocytometers (C-Chip, Biochrom).

**Clonogenic growth assays.** For clonogenic growth assays, RDES and TC-32 harboring shRNAs against *SOX6* were seeded at low density (200 cells) per well of a 12-well plate and grown for 21 days with renewal of Dox every 48 h. The colonies were counted in three technical replicates and the colony area was measured with the ImageJ Plugin 'Colony area'. The clonogenicity index was calculated by multiplying the counted colonies with the corresponding colony area.

**Sphere formation assays.** For the analysis of anchorage-independent growth, EwS cell lines RDES and TC-32 harboring Dox-inducible shRNAs against *SOX6*, were pre-treated with Dox for 48 h before seeding, whereas the *SOX6*-low expressing cell line A673 was pre-transfected with sipools against *SOX6* 48 h before seeding. Then, $1 \times 10^3$ cells/96-well were seeded in Costar Ultra-low attachment plates (Corning, Germany) for 12d. To maintain the *SOX6* knockdown during these 12d, 20 µl of fresh medium with/without Dox was added every 48 h to the RDES and TC-32 cells, whereas the *SOX6*-low expressing cell line A673 was re-transfected every 48 h with a sipool against *SOX6*. At day 12, wells were photographed and spheres larger than 500 µm in diameter were counted. The area was measured using ImageJ. The sphere volumes were calculated as follows: $V = 4/3 \times \pi \times r^3$. The sphere index was calculated by multiplying the counted colonies with the corresponding colony volume.

**3D bone model experiments.** For experiments in 3D in vitro mineralized bone models, TC-32/TR/shSOX6 EwS cells were additionally transduced with a vector containing the tdTomato gene that was constitutively expressed. To generate 3D in vitro bone models, human osteoblasts (2 strains, hOB, Sigma-Aldrich, USA and hOB Promocell, Germany) at a concentration of $1.5 \times 10^6$ cells/ml were mixed with $7.5 \times 10^4$ TC-32/TR/shSOX6 EwS cells per ml and embedded in a fibrin gel. The gel has been injected into u-shaped PMMA mask as and left to polymerize in a humidified incubator.

For proliferation experiments in 3D in vitro bone models TC-32/TR/shSOX6 were pre-treated with Dox for 48 h before seeding in treated samples. Samples were left in culture for 14 days with osteogenic medium (DMEM, 10% FBS, 2 mM L-glutamine, 1 mM sodium pyruvate, 10 mM HEPES, 1 U/ml penicillin and 1 µg/ml streptomycin (all from ThermoFisher, USA) supplemented with 0.01 µM dexamethasone, 10 mM β-glycerophosphate, 10 nM cholecalciferol, 150 µM L-ascorbic acid-2-phosphate (all from Sigma-Aldrich, USA), changing medium twice a week. For Elesclomol experiments, $7.5 \times 10^4$ TC-32/TR/shSOX6 cells were seeded within the 3D bone model and kept in culture for 36 h in normal medium. Afterwards, 10 nM Elesclomol was added to fresh normal medium and the 3D bone model was cultured for further 4 days. Mineralization of constructs was verified by staining hydroxyapatite deposits with Osteoimage® kit (Lonza, Switzerland). TC-32 volume fraction and expression of Ki67 (staining with anti-Ki67, ThermoFisher, USA) and cleaved caspase 3 (CC3) (staining with anti-CC3, ThermoFisher) were quantified on confocal microscopy image stacks using Image J plugins (Trainable Weka segmentation 3D and 3D geometric measure).

**Cell cycle analysis.** For cell cycle analysis, RDES and TC-32 cells harboring a Dox-inducible shRNA against *SOX6* were seeded at $4 \times 10^5$ cells per 10 cm dish and subsequently starved for 56 h. Stimulation of the cells was performed with 10% FCS for 20 h. Cells were fixed with ice-cold 70% ethanol, treated with 100 µg/ml RNAse (ThermoFisher, USA) and stained with 50 µg/ml propidium iodide (Sigma Aldrich). Analysis of the cell cycle was performed with BD Accuri C6 Cytometer (BD Biosciences) by counting at least $1 \times 10^5$ events. An example for the gating strategy is provided in Supplementary Fig. 7a.

**Annexin V staining.** For analysis of Annexin V positive cells, RDES and TC-32 cells harboring a Dox-inducible shRNA against *SOX6* were seeded at $3 \times 10^5$ cells per 10 cm dish and treated with 0.1 µg/ml Dox every 48 h for knockdown. After 96 h, cells were washed with PBS and cells were resuspended in $1 \times$ Annexin V buffer (BD Biosciences) with 5 µl of Annexin V and 5 µl PI solution for further 15

minutes. Analysis of Annexin V positivity was performed with BD Accuri C6 Cytometer (BD Biosciences) by counting at least $1 \times 10^5$ events. An example for the gating strategy is provided in Supplementary Fig. 7b.

**Oxidative stress detection via DCF-DA fluorescence**. For detection of oxidative stress, EwS cells harboring a Dox-inducible shRNA against *SOX6* were seeded at a density of $5 \times 10^4$ cells per 2 ml per 6-well and either directly treated for 96 h with Dox to induce the knockdown or treated with different reagents: 10 nM Elesclomol for 48 h, 0.01 mM Nac for 48 h, 0.1 mM Tiron (disodium 4,5-dihydroxy-1,3-benzenedisulfonate) for 48 h, 10 μM Menadione for 48 h, 1 μM DC_AC50 for 24 h or 1 μM BRD56491 for 24 h. For the knockdown of *TXNIP*, TC-32 cells were seeded at a density of $7 \times 10^4$ cells/2 ml per 6-well and reversely transfected with siRNA against *TXNIP* for 72 h. At the day of analysis, cells were incubated in their medium with 2.5 μM DCF-DA (ThermoFisher) for 30 min at 37 °C. Afterwards, cells were harvested and resuspended in PBS for flow cytometry analysis with Accuri C6 Cytometer (BD Biosciences). An example for the gating strategy is provided in Supplementary Fig. 7c.

**Mito-ROS detection via MitoSOX Red**. For detection of mitochondrial-ROS (mito-ROS) normalized to the cellular abundance of mitochondria, EwS cells were co-stained with MitoSOX Red and MitoTracker Green as recommended[68]. To this end, EwS cells were seeded at a density of $1.5 \times 10^4$ cells per 6-well and incubated with 200 nM MitoTracker Green in PBS for 30 min and afterwards co-stained with 5 μM MitoSOX Red for additional 15 min at 37 °C. Subsequently cells were washed twice with PBS, harvested and resuspended in PBS for flow cytometry analysis with an Accuri C6 Cytometer (BD Biosciences). An example for the gating strategy is provided in Supplementary Fig. 7d. To account for variations in relative fluorescence of MitoSOX Red and DCF-DA across experiments, data are shown as fold changes of the experimental groups over the respective controls that were normalized to 100% in each experiment.

**Gene expression and drug-response correlation**. To identify drugs whose efficacy correlates with *SOX6* expression in EwS cells, publicly available EwS cell line gene expression microarray data and drug-response values were downloaded from the EBI (E-MTAB-3610) and from www.cancerrxgene.org[31]. All CEL-files generated on Affymetrix Human Genome U219 arrays were simultaneously normalized using RMA[60] and a custom brainarray chip description file (v20, ENTREZG) yielding one optimized probe set for each gene[61]. For all drugs tested in EwS cell lines, the Pearson correlation coefficient and its significance between *SOX6* expression and IC50 values were calculated. Besides a high negative correlation coefficient and significance level ($P \le 0.05$), low IC50 values were chosen as criteria for selection of plausible and potentially relevant gene expression-drug-response dependencies.

**Drug-response assays and Elesclomol-treatment**. For Elesclomol-treatment, $2.5 \times 10^3$ cells of RDES and TC-32 with Dox-inducible *SOX6* knockdown or A673 cells with Dox-inducible *SOX6* overexpression as well as MSC-52 and SAOS-2 cell lines were seeded in wells of 96-well plates. Cells were pre-treated for 48 h with 0.1 μg/ml Dox to induce *SOX6* knockdown or with 1 μg/ml Dox to overexpress *SOX6*, before addition of Elesclomol (STA-4783) (Selleckchem, Germany). Different concentrations of Elesclomol ranging from 0.1 nM to 10 μM with/without Dox or 0.1–25 μM of Menadione or 50 μM–1 M DC_AC50/BRD56491 were added in a total volume of 100 μl per technical replicate for further 72 h.

In antioxidant experiments with N-acetyl cysteine (Nac) (Sigma-Aldrich) or Tiron (Sigma-Aldrich), cells were additionally treated with 0.01 mM Nac or 0.1 mM Tiron for 72 h. For the rescue experiments with $H_2O_2$, $2.5 \times 10^3$ cells/well were seeded in 96-well plates and *SOX6* knockdown was induced by addition of Dox. After 48 h, cells were either treated with Elesclomol (10 nM) or vehicle, and in addition with 30 mmol/l $H_2O_2$.

For drug-response assays in *TXNIP* silenced EwS cells, cells were first transfected with specific siRNAs directed against *TXNIP* (10 nmol) and after 48 h, Elesclomol was added to the culture media for 72 h. Similarly, for *TXNIP* rescue experiments, SOX6-silenced EwS cells with Cumate-inducible re-expression systems for *TXINP* were pre-treated with Cumate (5 μg/ml) before addition of Elesclomol. At the day of evaluation, Resazurin (16 μg/ml; Sigma-Aldrich) was added in order to measure cell viability. The relative IC50 concentrations were calculated using PRISM 5 (GraphPad Software Inc., CA, USA) and normalized to the respective controls.

**In vivo experiments in mice**. $3 \times 10^6$ EwS cells harboring a shRNA against *SOX6* were injected in a 1:1 mix of cells suspended in PBS with Geltrex Basement Membrane Mix (ThermoFisher) in the right flank of 10–12 weeks old NOD/Scid/gamma (NSG) mice. Tumor diameters were measured every second day with a caliper and tumor volume was calculated by the formula $L \times l^2/2$. When the tumors reached an average volume of 80 mm³, mice were randomized in two groups of which one was henceforth treated with 2 mg/ml Dox (BelaDox, Bela-pharm, Germany) dissolved in drinking water containing 5% sucrose (Sigma-Aldrich) to induce an in vivo knockdown (Dox (+)), whereas the other group only received 5% sucrose (control, Dox (−)). Once tumors of control groups reached an average

volume of 1,500 mm³, all mice of the experiment were sacrificed by cervical dislocation. Other humane endpoints were determined as follows: Ulcerated tumors, loss of 20% body weight, constant curved or crouched body posture, bloody diarrhea or rectal prolapse, abnormal breathing, severe dehydration, visible abdominal distention, obese Body Condition Scores (BCS), apathy, and self-isolation. For in vivo experiments using Elesclomol, EwS cells were subcutaneously injected in mice as described above. When the tumors reached an average volume of 80 mm³, mice were randomly distributed in equal groups and henceforth treated once per day intravenously (i.v.) with 5 mg/kg Elesclomol or vehicle (DMSO), interrupted for a two-day break on days 6 and 7 to allow mice to recover from the i.v. injections.

For analysis of EwS growth in bone, several bone injection models have been described for EwS[69,70]. In this study, EwS cells were orthotopically injected into the proximal tibial plateau of NSG mice[69]. To this end, mice were first analgo-anesthetized by i.p. injection with 0.5 mg/kg Medetomidine (Prodivet pharmaceuticals, Belgium), 5 mg/kg Midazolam (Roche Pharma, Germany) and 0.05 mg/kg Fentanyl (Dechra Veterinary Products, Germany) (per body weight). After disinfection of the injection site, $2 \times 10^5$ cells/20 μl were directly injected with a fine 28 G needle (Hamilton, USA) into the right proximal tibia. Thereafter, the initial analgo-anesthetics were antagonized with 1.2 mg/kg Naloxone (B. Braun Melsungen AG, Germany) and 2.5 mg/kg Atipamezole (Prodivet pharmaceuticals, Belgium) subcutaneously. For durable pain prophylaxis within the first day after intraosseous injection, mice were subsequently treated with 0.05 mg/kg Buprenorphine i.p. (Richter Pharma AG, Austria) (per body weight). At the first day after injection of tumor cells, mice were randomized in two groups of which one received henceforth 2 mg/ml Dox (BelaDox, Bela-pharm) dissolved in drinking water containing 5% sucrose (Sigma-Aldrich) to induce the *SOX6* knockdown (Dox (+)), whereas the other group only received 5% sucrose (control, Dox (−)). All tumor-bearing mice were sacrificed by cervical dislocation at the predefined experimental endpoint, when 40% of control tumors exceeded a volume of 1500 mm³ for subcutaneous injection models, or in the case of the orthotopic mouse model when the mice reached a humane endpoint as listed above or exhibited signs of limping at the injected leg. The tumors were extracted, small piece was snap frozen in liquid nitrogen for RNA isolation and the remaining tumor tissue was fixed in 4%-formalin and paraffin-embedded for immunohistology. Animal experiments were approved by the government of Upper Bavaria and conducted in accordance with ARRIVE guidelines, recommendations of the European Community (86/609/EEC), and United Kingdom Coordinating Committee on Cancer Research (UKCCCR) guidelines for the welfare and use of animals in cancer research.

**Patient-derived xenograft (PDX) models**. For the establishment of xenografts, tumor specimens from EwS patients were surgically removed and cut into individual pieces of 3–4 mm and subcutaneously transplanted into immunodeficient NOD/SCID mice. The sex of the recipient mice was chosen according to the donor patient. PDX tumor diameters were measured every second day with a caliper and tumor volumes were calculated by the formula $L \times l^2/2$. When the PDXs reached an average volume of 1500 mm³ (defined endpoint), mice were sacrificed by cervical dislocation. PDX tumors were extracted, fixed in 4% formalin and subsequently paraffin-embedded for immunohistology. All animal experiments were conducted in collaboration with Experimental Pharmacology and Oncology GmbH, Berlin Buch in accordance with the UKCCCR for the welfare and use of animals in cancer research, the German Animal Protection Law and had been approved by local authorities (LaGeSo, Berlin, Germany).

**Human samples and ethics approval**. Human tissue samples were analyzed at the Institute of Pathology of the LMU Munich (Germany) with approval of the institutional review board. All patients/guardians provided written informed consent. All analyses were approved by the ethics committee of the LMU Munich (approval no. 18-481 UE).

**Immunohistochemistry (IHC) and immunoreactivity scoring**. For IHC, 4-μm sections were cut and antigen retrieval was carried out by heat treatment with Target Retrieval Solution (S1699, Agilent Technologies, Germany) for SOX6 and Ki67, or Epitope Retrieval Solution pH6 (RE7113, Novocastra, Germany) for TXNIP staining. The slides were stained with either polyclonal anti-SOX6 antibody raised in rabbit (1:1600; HPA003908, Atlas Antibodies, Sweden) or with monoclonal anti-Ki67 raised in rabbit (1:200, 275R-15, Cell Marque/Sigma-Aldrich) or with a monoclonal anti-TXNIP antibody raised in rabbit (1:250; EPR14774, Abcam) for 60 min at RT, followed by a monoclonal secondary horseradish peroxidase (HRP)-coupled horse-anti-rabbit antibody (ImmPRESS Reagent Kit, MP-7401, Vector Laboratories, Germany). AEC-Plus (K3469, Agilent Technologies) for Ki67 and SOX6 staining or DAB + (K3468, Agilent Technologies) for TXNIP staining were used as chromogens. Samples were counterstained with hematoxylin (H-3401, Vector Laboratories). For cleaved caspase 3 staining, antigen retrieval was carried out by heat treatment with Target Retrieval Solution Citrate pH6 (S2369, Agilent Technologies). Slides were incubated with the polyclonal cleaved caspase 3 primary antibody (rabbit, 1:100; 9661, Cell Signaling, Frankfurt am Main, Germany) for 60 min at RT followed by ImmPRESS Reagent Kit. DAB + (K3468,

Agilent Technologies) was used as chromogen and hematoxylin for counterstaining. All formalin-fixed and paraffin-embedded (FFPE) xenografts of the EwS cell lines were stained with hematoxylin and eosin (H&E) to assess tissue integrity. Assessment of mitoses, necrosis and IHC markers was carried out on at least two sections per xenograft. Evaluation of immunoreactivity of SOX6 and TXNIP was carried out in analogy to scoring of hormone receptor Immune Reactive Score (IRS) ranging from 0–12. The percentage of cells with expression of the given antigen was scored and classified in five grades (grade 0 = 0–19%, grade 1 = 20–39%, grade 2 = 40–59%, grade 3 = 60–79% and grade 4 = 80–100%). In addition, the intensity of marker immunoreactivity was determined (grade 0 = none, grade 1 = low, grade 2 = moderate and grade 3 = strong). The product of these two grades defined the final IRS.

**Statistical analysis and software**. Statistical data analysis was performed using PRISM 5 (GraphPad Software Inc., Ca, USA) on the raw data. If not specified otherwise in the figure legends, comparison of two groups in functional in vitro experiments was carried out using a two-sided Mann–Whitney test. If not specified otherwise in the figure legends, data are presented as dot plots with horizontal bars representing means and whiskers representing the standard error of the mean (SEM). Sample size for all in vitro experiments were chosen empirically. For in vivo experiments, the sample size was predetermined using power calculations with $\beta = 0.8$ and $\alpha < 0.05$ based on preliminary data and in compliance with the 3R system (replacement, reduction, refinement). In Kaplan–Meier analyses of event-free survival, curves were calculated from all individual mice. Statistical differences between the groups were assessed by a Mantel–Haenszel test.

**Reporting summary**. Further information on research design is available in the Nature Research Reporting Summary linked to this paper.

## Data availability
The microarray data were deposited at the National Center for Biotechnology Information (NCBI) GEO database under the accession code GSE120576. The microarray, DNase-Seq and ChIP-Seq data referenced during the study are available in a public repository from the GEO website (https://www.ncbi.nlm.nih.gov/geo/); accession codes: GSE34620, GSE37371, GSE88826, GSE64686, GSM736570, and GSE61944. The source data underlying Figs. 1a–b, 2a–c, e–f, 3a–d, f–i, 4a–k, 5a–g, 6a–h and Supplementary Figs. 1a, 2a–c, 3a–j, 4a–c, 5a–c and 6a–c are provided as a Source Data file. All the other data supporting the findings of this study are available within the article and its supplementary information files and from the corresponding author upon reasonable request.

## Code availability
Custom code is available from the corresponding author upon reasonable request.

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

## Acknowledgements

The authors thank Dr. E. Tomazou and Dr. N. Sheffield for help in processing of the methylation data. We thank Dr. G. Leprivier, Dr. D. Surdez and Dr. S. von Karstedt for critical discussion, M. Melz for help in constructing TMAs, and A. Heier and A. Sendelhofert for expert technical assistance in immunohistological staining procedures. We thank Dr. V. R. Buchholz for support in cell cycle analysis. This work was mainly supported by a grant from the German Cancer Aid (DKH-111886). In addition, the laboratory of T.G.P.G. is supported by the 'Verein zur Förderung von Wissenschaft und Forschung an der Medizinischen Fakultät der LMU München' (WiFoMed), by LMU Munich's Institutional Strategy LMUexcellent within the framework of the German Excellence Initiative, the 'Mehr LEBEN für krebskranke Kinder—Bettina-Bräu-Stiftung', the Fritz-Thyssen Foundation (FTF-40.15.0.030MN), the Kind-Philipp-Foundation, the Matthias-Lackas Foundation, the Dr. Leopold and Carmen Ellinger Foundation, the Wilhelm Sander-Foundation (2016.167.1), the Barbara & Hubertus Trettner Foundation, the Dr. Rolf M. Schwiete Foundation, the Friedrich-Baur Foundation, the German Cancer Aid (DKH-70112257), the Gert und Susanna Mayer Foundation, the Barbara und Wilfried Mohr Foundation, and the Deutsche Forschungsgemeinschaft (DFG-391665916). M.D. was supported by a scholarship of the 'Deutsche Stiftung für junge Erwachsene mit Krebs', J.M. by a scholarship of the Kind-Philipp-Foundation, and T.L.B. H. by a scholarship from the German Cancer Aid. T.G.P.G., M.M. and C.A. were supported by the Swiss National Science Foundation (SNF 310030_179167). J.F.A. was supported by grants from the Cancer Prevention and Research Institute of Texas (RP120685) and the 1 Million 4 Anna Foundation. A.G.H. is supported by the Deutsche Forschungsgemeinschaft (DFG-398299703). A.G.H. is a participant in the BIH-Charité Clinical Scientist Program funded by the Charité—Universitätsmedizin Berlin and the Berlin Institute of Health.

## Author contributions

A.M. and T.G.P.G. conceived the study, wrote the paper, and drafted the figures and tables. J.S.G., M.F.O., and T.G.P.G. performed bioinformatic and statistical analyses. M.C. B. carried out gene expression analyses. A.M., M.M.L.K., J.L., and F.W. scored tissue-microarrays. S.O., F.C.A., and M.F.O. performed in vivo experiments. V.B. and A.G.H. characterized human PDX models. A.C. provided expert guidance in flow cytometric assessment of oxidative stress. M.C., C.A., and M.M. carried out experiments in 3D in vitro mineralized bone models. F.C.A., D.S., J.F.A., J.L., M.F.O., L.R.P., T.L.B.H., M.D., G.S., J.M., and S.S. contributed to experimental procedures. T.K. provided laboratory infrastructure and histological guidance. T.G.P.G. supervised the study and data analysis. All authors read and approved the final manuscript.

## Competing interests

The authors declare no competing interests.
