## [Peer Review File · Nature Communications]

Reviewers' comments:

Reviewer #1 (Remarks to the Author): Expert in transcriptomics

In this manuscript Marchetto, et al. identify SOX6 as a direct target for transcriptional regulation by the central fusion oncoprotein in Ewing sarcoma, EWSR1-FLI1, through an intronic enhancer. They demonstrate that SOX6 expression is necessary for the continued proliferation of Ewing sarcoma cells. Taking advantage of an existing dataset associating transcription with small molecule sensitivity, they demonstrate enhanced reactive oxygen species levels and sensitivity to elesclomol.

Overall, this is a well performed study, and the manuscript is well written. This study continues a longstanding effort to identify direct targets of EWSR1-FLI1 with the expectation that some of these targets will be critical for Ewing sarcoma survival and can be exploited therapeutically and ultimately inform the development of novel therapies. This project offers a variation on the traditional application of this approach. Rather than identifying a specific transcriptional target that itself could be therapeutically targeted, they identify that EWSR1-FLI1-regulated SOX6 creates a vulnerability that can be exploited. However, by and large, despite the identification of a small number of EWSR1-FLI1 targets, none has resulted in a novel therapy. This manuscript encompasses virtually the entire process of target identification through preclinical assessment and generates a link between Ewing sarcoma vulnerability through SOX6 regulation of TXNIP. The identification of SOX6-regulated enhanced ROS sensitivity is novel.

However, the several issues could be addressed to enhance the robustness of the described associations.

From both cell lines and primary tumor data, SOX6 is variably expressed. Is the effect of SOX6 silencing on cellular proliferation dependent on SOX6 levels across Ewing cell lines?

The majority of studies in this manuscript are performed in POE, RDES and TC-32 cell lines which demonstrate the most abundant SOX6 expression.

Similarly, demonstration of the association of GGAA repeat number with SOX6 levels primarily depends on analysis of those cell lines with the highest expression and lowest expression, with only one cell line with midrange SOX6 expression. Additional cell lines expressing midrange SOX6 should be included to further establish the variance in repeat number. It could be that length is not relevant, but variation reflects other regulatory factors (or that only longer repeats have increased expression).

There is a modest effect of reducing SOX6 levels by silencing on elesclomol sensitivity. Another known biological perturbation could be incorporated as a control against which to compare the anticipated magnitude of change.

Inclusion of another compound that induces reactive oxygen stress would be supportive of the role of elesclomol on ROS. Since the clinical development of elesclomol has apparently ended, it is not clear that there would be clinical path forward.

Figure 3c would benefit from inclusion of a low SOX6-expressing Ewing or non-Ewing cell line control.

Does SOX6 levels correspond with TXNIP RNA levels? To better support the model depicted in Figure 5g, the ability of ectopically expressed TXNIP in the context of SOX6 KD to reverse sensitivity to elesclomol should be assessed.

Minor

Y-axis labeling for Figure 4b could be clearer.

Colors using for Figure 2f scatter plot are hard to discriminate.

Reviewer #2 (Remarks to the Author): Expert in ROS and cancer

Please see attached file

Reviewer #3 (Remarks to the Author): Expert in mouse models

The present manuscript by Marchetto et al is generally an excellent study on sarcoma cells in vitro. However the in vivo model is obsolete and the manuscript and conclusions would be much stronger if the authors would use an orthotopic mouse model of Ewing's sarcoma, in particular a patient-derived orthotopic xenograft (PDOX) mouse model. This would make the manuscript clinically relevant (Oncotarget, 70, 47556, 2016); J Cell Biochem. 119. 967-972, 2017.

Marchetto et al. demonstrate that the prevalent EWSR1-FL11 translocation fusion protein in Ewing sarcoma is a de novo recombinant transcription factor that induces expression of SOX6 through interactions with GGAA microsatellite clusters. Compellingly, these GGAA microsatellites tend to be also expanded in this cancer type, and microsatellite repeat size correlates with SOX6 expression. SOX6, a developmental transcription factor, is shown to play a dominant role in this cancer by driving a proliferative and likely also morphogenic reprogramming. EWSR1-F11 has proven, to date, to be non-druggable. However, the SOX6-driven phenotype of EwS cells is associated with increased rates of conversion of administered DCFDA to fluorescent DCF. A drug-in-trial, Elesclomol, is also associated with DCFDA conversion, and in a screen of 264 drug candidates, Elesclomol was in the top 7 inhibitors of cancers with SOX6 overexpression and highly specific for these cells. Elesclomol was particularly effective against SOX6 overexpressing EwS cells. Elesclomol sensitivity was inhibited by siRNA-knockdown of either EWSR1-FL11 or SOX6, and this was shown both in culture and in xenograft models.

This is an enormously exciting and important study that should be moved toward publication. However, my previous paragraph very intentionally does not include the term "ROS". The submitted manuscript, including the title, implicates ROS based on misinterpretations of weak data. Once these unsupported interpretations are removed and a few other more minor issues are cleared up, this paper should be published with high priority.

Major concerns.

- 1) DCFDA was used to measure cellular ROS in this paper, as well as in the 2008 paper that reported ROS generation upon Elesclomol administration (PMID 18723479). However, it is now recognized that DCFDA has very complex intracellular chemistry. Importantly, it does not measure H₂O₂ nor does it accurately reflect ROS levels (PMID 22027063). DCFDA participates in both 1- and 2-electron reactions and, within cells it can function in radical-based redox-cycling, thereby itself artifactually promoting the production of super-physiological levels of ROS. Critically important, it can also accept electrons directly from cytochrome-C that leaks out of mitochondria during early apoptosis (PMID 22027063). In the submitted manuscript, DCF fluorescence was used as a measure of ROS, and the apparent increase in ROS was then inferred to induce apoptosis. Conversely, DCFDA reactivity could be a direct measure of CytC from apoptosis with no direct role for ROS. The authors should report the results (conversion of DCFDA) but not present their interpretation of this result (that the mechanism therefore involves induction of ROS) as a result, itself.
- 2) NAC prevents the DCFDA conversion in SOX6-overexpressing cells (this paper), as well as the DCFDA conversion in Elesclomol-treated cells (PMID 18723479). In both cases, the respective authors state that NAC is an ROS (or free radical) scavenger, and this is therefore given as evidence of a direct role of ROS in the SOX6-driven EwS phenotype (this paper) or in the mechanism of action of Elesclomol (PMID 18723479). However, it was shown two decades ago that, kinetically, NAC is neither a substantial ROS- nor free radical-scavenger (PMID 10468205 and others). How NAC actually does work has been a long-standing conundrum in the field. The field, however, is not well-served by propagating untenable conclusions. Recently, a kinetically tenable and data-supported

mechanism for the activity of NAC in similar situations was published (PMID 29429900), which should be taken into consideration here.

- 3) Elesclomol strongly chelates Cu(II) ions in culture medium, transport these into cells, and within the cell, it transports the Cu into mitochondria (PMID 22542443, 23707906). Whereas the Cu or Cu-liganded Eleclomol could be capturing electrons from respiration and generating ROS directly, it might instead be interfering with mitochondrial function in a ROS-independent manner, leading to release of CytC, which in turn could directly convert cytosolic DCFDA to DCF.
- 4) The data in this paper do not support the conclusion that SOX6 interferes with the antioxidant system. The only evidence given for this is a modest difference in Txnip expression which, itself, comes from cherry-picking the microarray data. Txnip expression differences were not independently verified at mRNA, protein, or enzyme activity levels. Despite its name (“Trx-inhibitory protein”) and the ability of Txnip to inhibit Txn1 redox activity in vitro, the idea that Txnip functions as a Txn inhibitor in vivo is controversial and is inconsistent with genetic data. Like the mechanistic implications for ROS in this paper, the implications for a role of the Txn system is both unsupported and unnecessary in this paper, and severely weakens the paper. This speculative conclusion should be removed from the Abstract, but might merit attention in the Discussion.
- 5) The microarray data provide the evidence that SOX6 is working in EwS as a transcription factor and not a splicing factor. This is a very important conclusion from the current study. However, the transcriptome dataset, itself, although rigorously analyzed, is poorly presented in the manuscript (Fig. 3b and Supplementary Tables 3 & 4). In the data presented, exceptionally soft threshold criteria were used to prioritize the data, resulting in a dataset with far too many genes to interpret (~4,000), which includes far too many false discovery artifacts. This brings interpretations of the data, e.g., concerning Txnip expression, into question. A cutoff of $P < 0.05$ was used rather than an FDR rate. $P < 0.05$ means that, for any given gene, there is up to a 5% chance that it appears by chance. Using the authors’ thresholds, nearly 4000 genes were found to differ significantly, $P < 0.05$. Therefore, up to 200 genes on the list were not truly affected, but instead were present only by chance. Next, a cutoff of $FC \pm 0.5 \log_2$ was used, which equals 1.4-fold up- or down-regulated. However, qPCR data presented in Supplementary Fig. S1 indicates that, in replicate analyses of cell cultures, expression levels of SOX6 vary over a ± 2 -fold range. With the regulator, itself, having this level of stochastic variance, how can such substantially smaller differences in downstream targets be considered important? From the volcano plot in Fig. 3B, one can predict that adjusting the threshold to a more stringent $1.0 \log_2$ (2-fold up or down) would likely trim >75% of the entries from the gene-list. This would help the authors and the reader to focus on the most important differences. Finally, Supplementary Tables 3 & 4 (the microarray data presentation) do not give quantitative hybridization values. It appears from the tables and the Methods description that no signal threshold was used for these datasets. At some pragmatic level, a threshold should be assigned. By using known expression levels of housekeeping genes, one could calculate that a certain hybridization signal represents an mRNA that is present at an average of 1 copy per cell. By excluding the likely enormous number of differentially represented genes whose highest mRNA level remained less than 1 molecule per cell, the authors could likely remove yet another large majority of data that is unlikely meaningful but, rather, is distracting and extraneous. Prudent use of

thresholds would allow the reader to focus on ~200 genes that reflect activities of the cells and ignore the other ~3800 genes that are not informative.

Minor points

- 1) Results section, 1st sentence (p. 3), is confusing. This should likely be rewritten to state that the authors "...took advantage of data from a well-curated set...".
- 2) Results p. 5 incorrectly states that SOX6 silencing induced a concordant up- or downregulation ($FC < -0.5$ and $FC > +0.5$). The Fig. legend (3B) indicates FC is shown in \log_2 , so this is actually $FC < -0.7$ and $FC > +1.4$.
- 3) The proliferation data (Supplementary Fig. 3d) should include cell cycle phase distributions in unsynchronized log-phase growth along with the current serum-restored synchronized populations. Serum-starvation is a severe stress that might, itself, impact cell cycle dynamics in ways that do not reflect the free-running cell cycle. If the ratios of G1, S and G2 in a free-running cycle correspond to the progression upon serum restoration, then they are robust.
- 4) It is inconceivable that, of all of the GGAA microsatellite clusters in cells that might (a) expand in with EwS transformation, (b) bind EWSR1-FL11, and (c) be near enough to a functional promoter to alter its activity, the only target of EWSR1-FL11 in EwS is SOX6. This reviewer is convinced by the data showing that SOX6 is, indeed, the phenotypically dominant target of EWSR1-FL11. However, the authors should consider in the Discussion possible roles of other direct EWSR1-FL11 targets in the global phenotype of these cells.
- 5) Elesclomol targets actively respiring mitochondria, and has diminished outcomes in suspected hypoxic conditions (PMID 23707906). It might be noteworthy to discuss the caveat of potentially poor therapeutic efficacy in large hypoxic tumors.

Step-by-step responses to the Reviewers

Authors: We thank all Reviewers for their time spent with our manuscript and their highly valuable and fair comments, which have been addressed in full.

Reviewer #1 remarks to the authors

In this manuscript, Marchetto et al. identify SOX6 as a direct target for transcriptional regulation by the central fusion oncoprotein in Ewing sarcoma, EWSR1-FLI1, through an intronic enhancer. They demonstrate that SOX6 expression is necessary for the continued proliferation of Ewing sarcoma cells. Taking advantage of an existing dataset associating transcription with small molecule sensitivity, they demonstrate enhanced reactive oxygen species levels and sensitivity to elesclomol.

Overall, this is a well performed study, and the manuscript is well written. This study continues a longstanding effort to identify direct targets of EWSR1-FLI1 with the expectation that some of these targets will be critical for Ewing sarcoma survival and can be exploited therapeutically and ultimately inform the development of novel therapies. This project offers a variation on the traditional application of this approach. Rather than identifying a specific transcriptional target that itself could be therapeutically targeted, they identify that EWSR1-FLI1-regulated SOX6 creates a vulnerability that can be exploited. However, by and large, despite the identification of a small number of EWSR1-FLI1 targets, none has resulted in a novel therapy. This manuscript encompasses virtually the entire process of target identification through preclinical assessment and generates a link between Ewing sarcoma vulnerability through SOX6 regulation of TXNIP. The identification of SOX6-regulated enhanced ROS sensitivity is novel.

Authors: We thank this Reviewer for pointing out the scientific quality of our manuscript and the novelty and relevance of our findings.

However, the several issues could be addressed to enhance the robustness of the described associations.

From both cell lines and primary tumor data, SOX6 is variably expressed. Is the effect of SOX6 silencing on cellular proliferation dependent on SOX6 levels across Ewing cell lines? The majority of studies in this manuscript are performed in POE, RDES and TC-32 cell lines which demonstrate the most abundant SOX6 expression.

Authors: We thank this Reviewer for this interesting question. To address this question, we have carried out transient knockdown experiments using pooled siRNAs (sipool) in two additional SOX6-low Ewing sarcoma (EwS) cell lines (A673 and EW7) and assessed their proliferation rate 96h after transfection. As can be seen from the revised Supplementary Fig. 3b, silencing of SOX6 in SOX6-low EwS cell lines had no significant impact on cell proliferation as compared to cell lines with high SOX6 levels, which may hint to the fact that only SOX6-

high cell lines depend in their proliferation on *SOX6* expression. These new data are now reported in the revised Results section.

Page 5, paragraph 4: “Interestingly, sipool-mediated silencing of *SOX6* had no effect on proliferation of A673 and EW7 cell lines, which only express low baseline levels of *SOX6* (**Supplementary Fig. 1a, Supplementary Fig. 3b**).”

Similarly, demonstration of the association of GGAA repeat number with *SOX6* levels primarily depends on analysis of those cell lines with the highest expression and lowest expression, with only one cell line with midrange *SOX6* expression. Additional cell lines expressing midrange *SOX6* should be included to further establish the variance in repeat number. It could be that length is not relevant, but variation reflects other regulatory factors (or that only longer repeats have increased expression).

Authors: We thank this Reviewer for these important remarks and have added two new cell lines (EW17 and ORS) with midrange *SOX6* expression (see Supplementary Fig. 1a). Adding these cell lines to the analysis corroborated our previously reported results, which became statistically even more significant ($P=0.016$). These additional results (now encompassing eight cell lines with high, midrange, or low *SOX6* expression) are shown in the revised Fig. 2f.

Furthermore, all cloned GGAA-microsatellites (mSats) were subjected to Sanger sequencing, which revealed that these mSats only varied in their numbers of consecutive GGAA-repeats, thus ruling out additional regulatory factors within the mSats. In addition, all parental cell lines were subjected to whole-genome sequencing, which confirmed that the cloned sequences flanking the *SOX6*-associated GGAA-mSat were entirely identical in all eight cell lines, thus ruling out additional regulatory elements outside of the mSat. These facts have now been explained in the revised Methods section.

Page 14, paragraph 2: “The presence of additional genetic variation in the cloned sequences apart from the *SOX6*-associated GGAA-mSat was ruled out by whole-genome sequencing of the parental cell lines and confirmatory Sanger sequencing of the cloned fragments.”

There is a modest effect of reducing *SOX6* levels by silencing on elesclomol sensitivity. Another known biological perturbation could be incorporated as a control against which to compare the anticipated magnitude of change.

Authors: We agree with this Reviewer concerning this aspect. As already shown in the initial Fig. 5b, treatment of EwS cells with the antioxidant Nac increased the IC50 of Elesclomol to a magnitude similar to that observed after knockdown of *SOX6* shown in Fig. 4f (~2–4-fold, depending on the cell line).

To corroborate these results, we carried out additional drug-response assays now treating the EwS cells with the antioxidant Tiron (Krishna CM *et al.*, 1992). As shown in the revised Fig. 5d, also treatment of EwS cells with Tiron significantly increased the IC50 of Elesclomol to a

magnitude similar to the *SOX6* knockdown. In addition, we carried out direct knockdown experiments for *TXNIP*, in which we observed a reduction of Elesclomol sensitivity of the EwS cells similar to the extent than *SOX6* knockdown. These new data are now reported in the revised Results section.

Page 8, paragraph 4: “To test whether elevated oxidative stress levels play a role in the capacity of Elesclomol to kill EwS cells, we carried out drug-response assays in the presence/absence of the antioxidants N-acetylcysteine (Nac) (Ezeriņa D *et al.*, 2018; Zafarullah M *et al.*, 2003) and Tiron (Krishna CM *et al.*, 1992). In both EwS cell lines, Nac- or Tiron-treatment resulted in significantly reduced conversion of DCF-DA to DCF (**Fig. 5c**) and decreased Elesclomol sensitivity (**Fig. 5d**), indicating that this drug may exert its pro-apoptotic effect in EwS to a certain degree via induction of oxidative stress over a tolerable threshold.”

Page 9, paragraph 6: “In accordance, direct knockdown of *TXNIP* reduced the sensitivity of EwS cells toward Elesclomol (**Fig. 6g**) to a similar extent than *SOX6* knockdown (**Fig. 4f**).”

Inclusion of another compound that induces reactive oxygen stress would be supportive of the role of elesclomol on ROS. Since the clinical development of elesclomol has apparently ended, it is not clear that there would be clinical path forward.

Authors: We fully agree with this Reviewer, that alternative compounds would be desirable. However, new data described in the responses to Reviewer #2 show that the phenotypical effect of Elesclomol on EwS cells may not only depend on cytoplasmic oxidative stress levels, but also on the induction of mitochondrial ROS, possibly due to unique functions of Elesclomol (see also question #3 of Reviewer #2).

Nevertheless, to address this Reviewer’s suggestion, we tested three different alternative substances, all of which are described in the literature to induce oxidative stress in cancer cells, namely Menadione, BRD56491, and DC_AC50. While we could not achieve a substantial induction of oxidative stress and subsequent cell death in EwS cells with BRD56491 and DC_AC50 at physiologically relevant concentrations (IC₅₀ much higher than 20 μM), Menadione was able to induce oxidative stress and cell death to a similar extent than Elesclomol. However, this effect was only achievable at relatively high IC₅₀ values (~10 μM), which are much higher than those of Elesclomol (~4 nM).

These results may point to a special role and potentially unique potency of Elesclomol in induction of oxidative stress in EwS and possibly other cancer cells. These new findings are now presented in the new Supplementary Figs. 5b-d and reported in the revised Discussion section.

Page 11, paragraph 2: “Since clinical development of Elesclomol has been discontinued, we explored whether other compounds could serve as alternatives. Interestingly, among three so-called oxidative stress inducers (BRD56491, DC_AC50, and Menadione), BRD56491 and DC_AC50 failed to induce oxidative stress and subsequent cell death at physiologically relevant concentrations in EwS cells (IC₅₀ >20 μM) (**Supplementary Fig. 5b**). In contrast, Menadione was able to induce oxidative stress (including mito-ROS) (**Supplementary Fig. 5c**)

and to decrease the viability of EwS cells expressing high *SOX6* levels, albeit only at 2,500-fold higher concentrations compared to Elesclomol (IC₅₀ ~10μM for Menadione vs. ~4 nM for Elesclomol) (**Supplementary Fig. 5d**), which may highlight the potency of Elesclomol.“

Figure 3c would benefit from inclusion of a low *SOX6*-expressing Ewing or non-Ewing cell line control.

Authors: We agree with this Reviewer and have repeated this experiment with the *SOX6*-low EwS cell line A673. Similar to the proliferation rates assessed in short-term knockdown experiments (96h) described above, long-term knockdown of *SOX6* (achieved by serial transfection with pooled siRNAs) did not change the sphere-formation capacity of A673 cells, suggesting that only *SOX6*-high EwS cell lines depend in their sphere-formation capacity on *SOX6*. These new data are now reported in the revised Results section and displayed in the revised Fig. 3c.

Page 5, paragraph 4: “In accordance, Dox-induced long-term *SOX6* knockdown significantly reduced the 2D clonogenic and 3D sphere-formation capacities of the highly *SOX6* expressing EwS cell lines RDES and TC-32 as compared to controls (Dox (-) and shCtrl) (**Fig. 3c**, **Supplementary Fig. 3c**), while *SOX6* silencing in the *SOX6*-low cell line A673 had no effect on sphere-formation capacity (**Fig. 3c**).”

Does *SOX6* levels correspond with *TXNIP* RNA levels?

Authors: We thank this Reviewer for this interesting question. As shown in the initial Fig. 5e (now revised Fig. 6b), we observed in two different EwS cell lines that the knockdown of *SOX6* mRNA is associated with a downregulation of the *TXNIP* mRNA expression. Thus, *SOX6* levels correspond to *TXNIP* mRNA levels. In addition, we now provide new data generated in 18 EwS cell lines profiled on Affymetrix Clariom D gene expression microarrays, which show a significant positive correlation of *SOX6* and *TXNIP*. These new data are now presented in the new Fig. 6a and reported in the revised Results section.

Page 9, paragraph 4: “We further investigated this finding by analyzing additional gene expression microarray data generated on the same microarray platform in 18 EwS cell lines (triplicate measurements per cell line). In this dataset, we found a significant positive correlation ($r_{\text{Pearson}}=0.559$, $P=0.016$) of the average *SOX6* and *TXNIP* mRNA expression levels across EwS cell lines (**Fig. 6a**).”

To better support the model depicted in Figure 5g, the ability of ectopically expressed *TXNIP* in the context of *SOX6* KD to reverse sensitivity to elesclomol should be assessed.

Authors: We thank this Reviewer for this important comment. To address this question, we have transduced a dox-inducible *SOX6* knockdown EwS cell line (TC-31/TR/sh*SOX6_2*) with

a cumate-inducible overexpression system that contains the full-length cDNA of *TXNIP*. These cells enabled to knockdown *SOX6* by addition of dox and to simultaneously restore *TXNIP* expression by addition of cumate. As shown in the new Fig. 6h, re-expression of *TXNIP* to ‘physiological’ levels rescued ~66% of the sensitivity of these EwS cells toward Elesclomol. These new data further support our model (now presented in Fig. 6i) that the EWSR1-FLI1-induced strong upregulation of *SOX6* contributes, at least in part, via upregulation of *TXNIP* and oxidative stress levels to the high sensitivity of EwS cells toward the oxidative stress-inducing drug Elesclomol. These new results are reported in the revised Results section:

Page 9, paragraphs 6 and 7: “Even more strikingly, inducible re-expression of *TXNIP* in *SOX6* silenced EwS cells onto ‘physiological’ levels was sufficient to rescue ~66% of the sensitivity of these EwS cells toward Elesclomol (**Fig. 6h**).

Taken together, these data provide evidence that the EWSR1-FLI1-induced strong upregulation of *SOX6* contributes, at least in part, via upregulation of *TXNIP* and oxidative stress levels to the high sensitivity of EwS cells toward the oxidative stress-inducing drug Elesclomol (**Fig. 6i**).”

Minor

Y-axis labeling for Figure 4b could be clearer.

Authors: We apologize for this confusion and clarified the graph and its legend by now plotting the data in natural scale.

Colors using for Figure 2f scatter plot are hard to discriminate.

Authors: We updated the color code of Fig. 2f, which should be now better to discriminate.

Reviewer #2 remarks to the authors

Marchetto et al. demonstrate that the prevalent EWSR1-FL11 translocation fusion protein in Ewing sarcoma is a *de novo* recombinant transcription factor that induces expression of *SOX6* through interactions with GGAA microsatellite clusters.

Compellingly, these GGAA microsatellites tend to be also expanded in this cancer type, and microsatellite repeat size correlates with *SOX6* expression. *SOX6*, a developmental transcription factor, is shown to play a dominant role in this cancer by driving a proliferative and likely also morphogenic reprogramming. EWSR1-F11 has proven, to date, to be non-druggable. However, the *SOX6*-driven phenotype of EwS cells is associated with increased rates of conversion of administered DCF-DA to fluorescent DCF. A drug-in-trial, Elesclomol, is also associated with DCF-DA conversion, and in a screen of 264 drug candidates, Elesclomol was in the top 7 inhibitors of cancers with *SOX6* overexpression and highly specific for these cells. Elesclomol was particularly effective against *SOX6* overexpressing EwS cells.

Elesclomol sensitivity was inhibited by siRNA-knockdown of either EWSR1-FL11 or SOX6, and this was shown both in culture and in xenograft models. This is an enormously exciting and important study that should be moved toward publication.

Authors: We thank this Reviewer for pointing out the relevance and importance of our study, and for the recommendation that it should be moved toward publication.

However, my previous paragraph very intentionally does not include the term “ROS”. The submitted manuscript, including the title, implicates ROS based on misinterpretations of weak data. Once these unsupported interpretations are removed and a few other more minor issues are cleared up, this paper should be published with high priority.

Authors: We thank this Reviewer for pointing out that our manuscript should be published with high priority. We are very grateful for the insightful comments concerning the term ROS. Our manuscripts including its title has been entirely revised according to these highly valuable recommendations (see below).

Major concerns.

1) DCFDA was used to measure cellular ROS in this paper, as well as in the 2008 paper that reported ROS generation upon Elesclomol administration (PMID 18723479). However, it is now recognized that DCFDA has very complex intracellular chemistry. Importantly, it does not measure H₂O₂ nor does it accurately reflect ROS levels (PMID 22027063). DCFDA participates in both 1- and 2-electron reactions and, within cells it can function in radical-based redox cycling, thereby itself artifactually promoting the production of super-physiological levels of ROS. Critically important, it can also accept electrons directly from cytochrome-C that leaks out of mitochondria during early apoptosis (PMID 22027063).

In the submitted manuscript, DCF fluorescence was used as a measure of ROS, and the apparent increase in ROS was then inferred to induce apoptosis. Conversely, DCFDA reactivity could be a direct measure of CytC from apoptosis with no direct role for ROS. The authors should report the results (conversion of DCFDA) but not present their interpretation of this result (that the mechanism therefore involves induction of ROS) as a result, itself.

Authors: We thank this Reviewer for this insightful comment. To address this concern, we consequently refer to DCF-DA to DCF conversion when describing the corresponding experiments. To corroborate our results, we now use additional antioxidants (see reply to question #2). Furthermore, we now assessed the induction of mitochondrial ROS (mito-ROS) by Elesclomol as well as the modulation of mito-ROS by knockdown of *SOX6* in EwS cells as described below (see reply to question #3).

2) NAC prevents the DCFDA conversion in *SOX6*-overexpressing cells (this paper), as well as the DCFDA conversion in Elesclomol-treated cells (PMID 18723479). In both cases, the respective authors state that NAC is an ROS (or free radical) scavenger, and this is therefore

given as evidence of a direct role of ROS in the SOX6-driven EwS phenotype (this paper) or in the mechanism of action of Elesclomol (PMID 18723479). However, it was shown two decades ago that, kinetically, NAC is neither a substantial ROS- nor free radical-scavenger (PMID 10468205 and others). How NAC actually does work has been a long-standing conundrum in the field. The field, however, is not well-served by propagating untenable conclusions. Recently, a kinetically tenable and data-supported mechanism for the activity of NAC in similar situations was published (PMID 29429900), which should be taken into consideration here.

Authors: We thank this Reviewer for this insightful comment. To address this concern, we have modified the description of Nac and cite the suggested paper by Ezeriņa D *et al.* 2018 ('N-Acetyl Cysteine Functions as a Fast-Acting Antioxidant by Triggering Intracellular H₂S and Sulfane Sulfur Production') when introducing the antioxidant Nac in the revised Results section.

In addition, we tested whether Nac-treatment can reduce the conversion of DCF-DA to DCF in two independent EwS cell lines (TC-32 and RDES). As shown in the revised Fig. 5c, treatment of both EwS cell lines with Nac reduced the conversion rate of DCF-DA to DCF, suggesting that Nac may function as an antioxidant in these cells.

To further confirm the results obtained by Nac-treatment and to comply with a question of Reviewer #1, we repeated these experiments using another antioxidant (Tiron). Similar to Nac-treatment, also Tiron-treatment reduced the conversion rate of DCF-DA to DCF in both EwS cell lines. Strikingly, Nac- as well as Tiron-treatment of both EwS cell lines reduced their sensitivity toward Elesclomol. These new findings are now displayed in the revised Figs. 5c,d and reported in the revised Results section.

Page 8, paragraph 4: "To test whether elevated oxidative stress levels play a role in the capacity of Elesclomol to kill EwS cells, we carried out drug-response assays in the presence/absence of the antioxidants N-acetylcysteine (Nac) (Ezeriņa D *et al.*, 2018; Zafarullah M *et al.*, 2003) and Tiron (Krishna CM *et al.*, 1992). In both EwS cell lines, Nac- or Tiron-treatment resulted in significantly reduced conversion of DCF-DA to DCF (**Fig. 5c**) and decreased Elesclomol sensitivity (**Fig. 5d**), indicating that this drug may exert its pro-apoptotic effect in EwS to a certain degree via induction of oxidative stress over a tolerable threshold."

3) Elesclomol strongly chelates Cu(II) ions in culture medium, transports these into cells, and within the cell, it transports the Cu into mitochondria (PMID 22542443, 23707906). Whereas the Cu or Cu-liganded Elesclomol could be capturing electrons from respiration and generating ROS directly, it might instead be interfering with mitochondrial function in a ROS-independent manner, leading to release of CytC, which in turn could directly convert cytosolic DCFDA to DCF.

Authors: We thank this Reviewer for this insightful comment. We now mention these important aspects in the revised Results and Discussion section with reference to the suggested papers. Furthermore, we address this possibility experimentally using MitoSOX Red and MitoTracker Green stains in EwS cells upon Elesclomol treatment. In fact, when staining EwS cells treated

with Elesclomol with MitoSOX Red and MitoTracker Green, we observed a strong increase of MitoSOX Red fluorescence while we did not observe a change in mitochondrial mass (MitoTracker Green). These new results are displayed in the revised Fig. 5b and reported in the revised Results section. In addition, we cite the suggested references.

Page 8, paragraph 3: “Since Elesclomol may be able to interact with the electron transport chain and to strongly induce intra-mitochondrial reactive oxygen species (mito-ROS) (Nagai M *et al.*, 2012; Yadav AA *et al.*, 2013), we stained EwS cells after Elesclomol treatment with MitoSOX Red, which can detect mitochondrial superoxide anions (De Biasi S *et al.*, 2016; Ikeda K *et al.*, 2019), and normalized the resulting MitoSOX Red fluorescence signal to the mitochondrial mass as determined by MitoTracker Green stains (De Biasi S *et al.*, 2016; Ikeda K *et al.*, 2019). Indeed, we found that Elesclomol treatment significantly induced the relative fluorescence signal for MitoSOX Red (**Fig. 5b**), which may suggest that at least part of the oxidative stress induced by Elesclomol is derived from mito-ROS.”

4) The data in this paper do not support the conclusion that SOX6 interferes with the antioxidant system. The only evidence given for this is a modest difference in Txnip expression which, itself, comes from cherry-picking the microarray data. Txnip expression differences were not independently verified at mRNA, protein, or enzyme activity levels.

Authors: We thank this Reviewer for this comment. In contrast to the statement of this Reviewer, the downregulation of TXNIP, which was discovered in the microarray experiment, was verified in multiple independent experiments at the mRNA as well as at the protein level *in vitro*.

Indeed, it may have escaped to this Reviewer’s notice that these multiple independent validation experiments were already shown in Fig. 5e of our initial manuscript (now Fig. 6b).

As shown in the current Fig. 6b (formerly Fig. 5e), we found – depending on the levels of SOX6 suppression – a strong downregulation of TXNIP up to ~10% of controls, which we do not consider as being ‘modest’.

Moreover, in our microarray discovery experiment, *TXNIP* ranged among the most strongly downregulated genes upon *SOX6* knockdown in two independent EwS cell lines, which was even more pronounced when applying the more stringent filtering steps of the microarray data as proposed by this Reviewer (see comments on question #5 below as well as the revised Fig. 3b and revised Supplementary Table 3). Indeed, *TXNIP* is now the second most strongly downregulated gene after knockdown of *SOX6* in both EwS cell lines (mean log₂ FC –2.34). Since *TXNIP* is strongly downregulated upon *SOX6* knockdown, which could be validated in multiple independent experiments, we believe that this provides sufficient justification to have further investigated the role of *TXNIP* in the context of *SOX6*, oxidative stress, and Elesclomol sensitivity in EwS.

In addition, and in line with the findings from our initial validation experiments, we now provide new data generated in 18 EwS cell lines profiled by Affymetrix Clariom D gene

expression microarrays (triplicates per cell line) in which we found a significant ($P=0.016$) positive correlation of *TXNIP* and *SOX6* mRNA levels across EwS cell lines (revised Fig. 6a). To further validate the regulatory relationship between *SOX6* and *TXNIP*, we analyzed RNA from our EwS xenografts by qRT-PCR and found that *TXNIP* mRNA levels are also strongly downregulated after *SOX6* silencing *in vivo*. Similarly, we observed a strong downregulation of *TXNIP* on the protein level in the same xenografts as evidenced by IHC. These new results are now presented in the revised Figs. 6a-c and reported in the revised Results section.

Page 9, paragraphs 3–5: “To explore possible links between *SOX6*, oxidative stress, and Elesclomol sensitivity in EwS cells, we reassessed our microarray data obtained from EwS cells with/without knockdown of *SOX6*. Of note, *TXNIP* (thioredoxin interacting protein) – a key inhibitor of the thioredoxin antioxidant system (Burke-Gaffney A *et al.*, 2005; Hwang J *et al.*, 2014) – was the second most strongly downregulated gene after *SOX6* silencing in this discovery experiment (**Supplementary Table 3**).

We further investigated this finding by analyzing additional microarray gene expression data generated on the same microarray platform in 18 EwS cell lines (triplicate measurements per cell line). In this dataset, we found a significant positive correlation ($r_{\text{Pearson}}=0.559$, $P=0.016$) of the average *SOX6* and *TXNIP* mRNA expression levels across EwS cell lines (**Fig. 6a**).

To corroborate the link between *SOX6* and *TXNIP*, we carried out multiple independent *SOX6* knockdown experiments in which we detected a consistent downregulation of *TXNIP* at the mRNA and protein level *in vitro* (**Fig. 6b**). In addition, we found a significant downregulation of *TXNIP* *in vivo* by immunohistochemistry in xenografts with *SOX6* knockdown (**Fig. 6c**).”

Despite its name (“Trx-inhibitory protein”) and the ability of *Txnip* to inhibit *Txn1* redox activity *in vitro*, the idea that *Txnip* functions as a *Txn* inhibitor *in vivo* is controversial and is inconsistent with genetic data. Like the mechanistic implications for ROS in this paper, the implications for a role of the *Txn* system is both unsupported and unnecessary in this paper, and severely weakens the paper. This speculative conclusion should be removed from the Abstract, but might merit attention in the Discussion.

Authors: We thank this Reviewer for sharing his/her opinion, However, we politely disagree with this Reviewer because we find that in light of the new *in vitro* and *in vivo* data described above, which were generated in multiple independent experiments, the link between *SOX6* and *TXNIP* has been further established.

Moreover, we have initially shown (initial Fig. 5f) that the direct knockdown of *TXNIP* in EwS cells reduces the conversion rate of DCF-DA to DCF, indicating that *TXNIP* may promote oxidative stress in EwS (now Figs. 6d,e). These initial findings are now further supported by new data demonstrating that the direct knockdown of *TXNIP* in EwS cells reduces the relative levels of MitoSOX Red fluorescence. In addition, we now show that the direct knockdown of *TXNIP* in EwS cells decreases their sensitivity toward Elesclomol to a similar extent than *SOX6* knockdown. These new data are now reported in the new Figs. 6d–g and the revised Results section.

Page 9, paragraph 6: “Interestingly, direct siRNA-mediated knockdown of *TXNIP* in EwS cells (**Fig. 6d**) significantly reduced the conversion rate of DCF-DA to DCF (**Fig. 6e**), and the relative levels of MitoSOX Red fluorescence (**Fig. 6f**). In accordance, direct knockdown of *TXNIP* reduced the sensitivity of EwS cells toward Elesclomol (**Fig. 6g**) to a similar extent than *SOX6* knockdown (**Fig. 4f**).”

Furthermore, we now show new results from *TXNIP* rescue experiments (as recommended by Reviewer #1). These experiments demonstrated that restoration of *TXNIP* in *SOX6* silenced EwS cells onto ‘physiological’ levels can rescue ~66% of their sensitivity toward Elesclomol (Fig. 6h). These new data are now provided in the revised Results section.

Page 9, paragraph 6: “Even more strikingly, inducible re-expression of *TXNIP* in *SOX6* silenced EwS cells onto ‘physiological’ levels was sufficient to rescue ~66% of the sensitivity of these EwS cells toward Elesclomol (**Fig. 6h**).”

However, to accommodate this Reviewer’s concern, we have modified the Abstract as well as the concluding remark of the corresponding Results section, and added a critical discussion on the controversial role of *TXNIP* in the antioxidant system to the revised Discussion section. Moreover, we now indicate that also other factors than *TXNIP* may play a role in the *SOX6*-mediated Elesclomol sensitivity of EwS cells.

Page 9, paragraph 7: “Taken together, these data provide evidence that the EWSR1-FLI1-induced strong upregulation of *SOX6* contributes, at least in part, via upregulation of *TXNIP* and oxidative stress levels to the high sensitivity of EwS cells toward the oxidative stress-inducing drug Elesclomol (**Fig. 6i**)”.

Page 12, paragraph 1–2: “The elevated intracellular oxidative stress and mito-ROS levels as well as the associated hyper-sensitivity of *SOX6* high-expressing EwS cells toward Elesclomol can be explained, at least in part, by the *SOX6*-mediated upregulation of *TXNIP*. In fact, *TXNIP* knockdown in EwS cells was associated with decreased conversion rates of DCF-DA to DCF and lower levels of relative MitoSOX Red fluorescence, which might be caused by its potential inhibitory function on the thioredoxin (TRX) antioxidant system that plays an essential role in buffering intracellular oxidative stress (Burke-Gaffney A *et al.*, 2005; Hwang J *et al.*, 2014). In our rescue experiments, we noted that restoration of *TXNIP* expression (**Fig. 6h**) in *SOX6* silenced EwS cells could rescue ~66% of the Elesclomol sensitivity. These data indicate that *TXNIP* is an important downstream mediator of the *SOX6*-induced Elesclomol sensitivity of EwS cells, but also suggest that other factors may play a role.

Besides, the role of *TXNIP* in EwS may extend beyond the TRX antioxidant system, since *TXNIP* has also been reported to impact on glucose and lipid metabolism (Alhawiti NM *et al.*, 2017). While the potentially pro-proliferative role of *TXNIP* in EwS remains to be fully elucidated, it should be noted that in other cancers including hepatocellular, breast, and bladder carcinoma as well as leukemia, *TXNIP* has been reported to act as a tumor suppressor (Zhou J

and Chng WJ, 2013), indicating that this protein may, like SOX6, either promote or inhibit tumor growth depending on the cellular context.”

5) The microarray data provide the evidence that SOX6 is working in EwS as a transcription factor and not a splicing factor. This is a very important conclusion from the current study.

Authors: We thank this Reviewer for pointing out the high importance of this finding.

However, the transcriptome dataset, itself, although rigorously analyzed, is poorly presented in the manuscript (Fig. 3b and Supplementary Tables 3 & 4). In the data presented, exceptionally soft threshold criteria were used to prioritize the data, resulting in a dataset with far too many genes to interpret (~4,000), which includes far too many false discovery artifacts. This brings interpretations of the data, e.g., concerning Txnip expression, into question. A cutoff of $P < 0.05$ was used rather than an FDR rate. $P < 0.05$ means that, for any given gene, there is up to a 5% chance that it appears by chance. Using the authors' thresholds, nearly 4000 genes were found to differ significantly, $P < 0.05$. Therefore, up to 200 genes on the list were not truly affected, but instead were present only by chance. Next, a cutoff of $FC \pm 0.5 \log_2$ was used, which equals 1.4-fold up- or down-regulated. However, qPCR data presented in Supplementary Fig. S1 indicates that, in replicate analyses of cell cultures, expression levels of SOX6 vary over a ± 2 -fold range. With the regulator, itself, having this level of stochastic variance, how can such substantially smaller differences in downstream targets be considered important? From the volcano plot in Fig. 3B, one can predict that adjusting the threshold to a more stringent $1.0 \log_2$ (2-fold up or down) would likely trim >75% of the entries from the gene-list. This would help the authors and the reader to focus on the most important differences. Finally, Supplementary Tables 3 & 4 (the microarray data presentation) do not give quantitative hybridization values. It appears from the tables and the Methods description that no signal threshold was used for these datasets. At some pragmatic level, a threshold should be assigned. By using known expression levels of housekeeping genes, one could calculate that a certain hybridization signal represents an mRNA that is present at an average of 1 copy per cell. By excluding the likely enormous number of differentially represented genes whose highest mRNA level remained less than 1 molecule per cell, the authors could likely remove yet another large majority of data that is unlikely meaningful but, rather, is distracting and extraneous. Prudent use of thresholds would allow the reader to focus on ~200 genes that reflect activities of the cells and ignore the other ~3800 genes that are not informative.

Authors: We thank this Reviewer for highlighting that our microarray data have been rigorously analyzed. However, we agree with this Reviewer and have applied more stringent filtering steps of the microarray data as suggested.

Since the new Affymetrix microarray type (Clariom D), which was used for the current study, differs in its technology from previous generations of dual color Affymetrix microarrays, we used a slightly different method to identify minimally or virtually not expressed genes represented on this new microarray type. To this end, we excluded all genes from further

analysis, which showed gene expression signals as low as or just minimally higher than that observed for *ERG* (mean log₂ expression signal of 6.05), which is known to be virtually not expressed in EWSR1-FLI1 positive EwS cell lines (Crompton B *et al.*, 2014). By applying a pragmatic signal threshold of 7 (log₂), this filtering process yielded – as anticipated by this Reviewer – a largely reduced list of genes for subsequent analysis, which now only contained at least minimally expressed genes.

Using this filtered gene list, we repeated our gene set enrichment analysis (GSEA), which entirely confirmed the strong depletion of gene sets involved in proliferation and cell cycle progression in EwS cells upon knockdown of *SOX6*.

Next, as suggested by this Reviewer, we only considered genes as being potential differentially expressed genes (DEGs) in case of a minimum absolute log₂ signal fold change of 1. Moreover, we calculated the false discovery rate (FDR) estimate for each gene using the R package qvalue from BioConductor. The updated volcano plot, which is now plotting the FDR estimate against the mean gene expression FC is now presented in the revised Fig. 3b. The Results and Methods sections have been revised accordingly:

Page 5, paragraph 3: “In fact, *SOX6* silencing induced a concordant up- or downregulation (min. absolute log₂ FC of 1) of 54 and 499 genes, respectively, across shRNAs and cell lines (**Supplementary Table 3**).”

Page 16, paragraph 3: “First, normalized gene expression signal was log₂ transformed. To avoid false discovery artifacts due to the detection of only minimally expressed genes, we excluded all genes with a lower or just minimally higher gene expression signal than that observed for *ERG* (mean log₂ expression signal of 6.05), which is known to be virtually not expressed in EWSR1-FLI1 positive EwS cell lines (Crompton B *et al.*, 2014). Accordingly, only genes with mean log₂ expression signals (w/o Dox condition) of at least 7 were further analyzed. The FCs of the shControl samples and both specific shRNAs were calculated for each cell line separately. Then the FCs observed in the respective shControl samples were subtracted from those seen in the shSOX6 samples, which yielded the FCs for each specific shRNAs in each cell line. Then these FCs were averaged to obtain the mean FC per gene across shRNAs and cell lines. As the log₂ FC for *SOX6* was –1.486, we assumed that downstream targets should also be strongly regulated. Consequently, only genes with a minimum absolute log₂ FC of 1 were considered as DEGs. Additionally, the false discovery rate (FDR) was estimated for each gene using the R package qvalue from BioConductor (Storey J *et al.*, 2019).”

Interestingly, after applying these more stringent filtering process, *TXNIP* was the second most strongly downregulated gene (Supplementary Table 3) after knockdown of *SOX6* with a mean log₂ FC of –2.34 across both EwS cell lines, which further supports the link between *SOX6* and *TXNIP* that has been validated in multiple independent experiments at the mRNA and protein level *in vitro* and *in vivo* (see also comments above).

Minor points

1) Results section, 1st sentence (p. 3), is confusing. This should likely be rewritten to state that the authors "...took advantage of data from a well-curated set...".

Authors: This has been corrected.

Page 3, paragraph 5: "To explore the expression pattern of SOX6, we took advantage of a well-curated set of >750 DNA microarrays, which we established previously (Baldauf MC et al., 2018a, 2018b), comprising 18 representative normal tissues types and 10 cancer entities."

2) Results p. 5 incorrectly states that SOX6 silencing induced a concordant up- or downregulation ($FC < -0.5$ and $FC > +0.5$). The Fig. legend (3B) indicates FC is shown in log2, so this is actually $FC < -0.7$ and $FC > +1.4$.

Authors: This has been corrected and adapted to the new thresholds (see above).

3) The proliferation data (Supplementary Fig. 3d) should include cell cycle phase distributions in unsynchronized log-phase growth along with the current serum restored synchronized populations. Serum-starvation is a severe stress that might, itself, impact cell cycle dynamics in ways that do not reflect the freerunning cell cycle. If the ratios of G1, S and G2 in a free-running cycle correspond to the progression upon serum restoration, then they are robust.

Authors: We thank this Reviewer for this interesting remark. However, our cell cycle analysis was fully controlled by adding a control group with a control shRNA. In addition, our multiple independent *in vivo* experiments consistently showed that knockdown of SOX6 leads to reduced mitotic counts (Fig. 3f) and nuclear Ki67 expression (Supplementary Fig. 3g), which is in line with our strong depletion of cell cycle signatures identified by transcriptome profiling in freerunning (i.e. non-serum-starved) SOX6 silenced EwS cell lines (Fig. 3b).

Further, our findings are supported by data from the literature showing that SOX6 promotes proliferation of osteogenic progenitor cells (Akiyama H *et al.*, 2002; Lefebvre V *et al.*, 1998 & 2005; Smits P *et al.*, 2004). Moreover, we now provide new data showing a significant positive correlation of SOX6 protein expression, Ki67 expression and tumor growth in eight independent EwS PDX models *in vivo* (see below and new Fig. 3h), and SOX6 knockdown in EwS cells in mineralized bone model exhibit reduced Ki67 expression levels as well (new Supplementary Fig. 3e). Thus, we believe that our results shown in the revised Supplementary Fig. 3d are not caused by artifacts produced by serum-starvation, which is routinely applied in cell cycle analyses.

However, to accommodate this Reviewer's concern, we now added to the cell cycle analysis (Supplementary Fig. 3d) the fractions of cells in sub-G1 phase. These new data demonstrate that despite serum-starvation, the number of dead cells (sub-G1) was exceedingly low and equal across groups (Dox (-) vs. Dox (+)), thus further ruling out that serum-starvation may have excessively stressed our cells.

4) It is inconceivable that, of all of the GGAA microsatellite clusters in cells that might (a) expand in with EwS transformation, (b) bind EWSR1-FLI1, and (c) be near enough to a functional promoter to alter its activity, the only target of EWSR1-FLI1 in EwS is SOX6. This reviewer is convinced by the data showing that SOX6 is, indeed, the phenotypically dominant target of EWSR1-FLI1. However, the authors should consider in the Discussion possible roles of other direct EWSR1-FLI1 targets in the global phenotype of these cells.

Authors: We thank this Reviewer for this interesting suggestion and added two paragraphs to the revised Discussion section on previously identified direct EWSR1-FLI1 target genes driven by GGAA-microsatellites (mSat) and their impact on the phenotype of EwS cells.

Page 10, paragraph 2: “These findings are in line with recent observations for other direct EWSR1-FLI1 target genes such as *EGR2*, *MYBL2*, and *NROB1* whose variable expression in EwS tumors is caused by inter-individual differences in GGAA-repeat numbers of the corresponding enhancer-like GGAA-mSat (Grünewald TG *et al.*, 2015; Monument MJ *et al.*, 2016; Musa J *et al.*, 2019).

Page 10, paragraph 3: “In this regard, it is interesting to note that other direct EWSR1-FLI1 target genes, such as *EGR2*, *MYBL2*, and *NROB1*, which are driven by GGAA-mSats, are also critical mediators of EwS tumorigenesis (Grünewald TG *et al.*, 2015; Kinsey M *et al.*, 2006; Musa J *et al.*, 2019).”

5) Elesclomol targets actively respiring mitochondria, and has diminished outcomes in suspected hypoxic conditions (PMID 23707906). It might be noteworthy to discuss the caveat of potentially poor therapeutic efficacy in large hypoxic tumors.

Authors: We thank this Reviewer for this important remark. We have mentioned this fact in the revised Discussion section and cited the suggested reference.

Page 12, paragraph 3: “However, as Elesclomol targets actively respiring mitochondria, it may have less efficacy in hypoxic conditions (Yadav AA *et al.*, 2013), which may be the case in large hypoxic tumors.”

Reviewer #3 remarks to the authors

The present manuscript by Marchetto et al is generally an excellent study on sarcoma cells in vitro.

Authors: We thank this Reviewer by pointing out the excellence of our *in vitro* experiments.

However the *in vivo* model is obsolete and the manuscript and conclusions would be much stronger if the authors would use an orthotopic mouse model of Ewing’s sarcoma, in particular

a patient-derived orthotopic xenograft (PDOX) mouse model. This would make the manuscript clinically relevant (Oncotarget, 70, 47556, 2016); J Cell Biochem. 119. 967-972, 2017.

Authors: We thank this Reviewer for sharing his/her opinion. However, since the cell of origin of EwS has not been identified yet, and since all attempts to generate genetically engineered mouse models for this disease, thus far, have either failed or not led to *bona fide* models (for review see Minas TZ *et al.* 2016 Oncotarget & Grünewald TG *et al.* 2018 Nature Reviews Disease Primers), xenotransplantation models still constitute the mainstay of *in vivo* EwS research, which have been extensively carried out in this study and which yielded consistent results across cell lines and across shRNAs used, and which were in full agreement with our *in vitro* findings.

However, we agree with this Reviewer that more sophisticated models could further strengthen the conclusions of our manuscript. Hence, we have replicated key experiments in a wide spectrum of additional models including 3D advanced mineralized bone models, orthotopic xenotransplantation models and PDX models, all of which fully confirmed our previous conclusions (as explained below in detail).

The two papers mentioned by this Reviewer, which were both published by the company AntiCancer Inc. that was founded and headed by Dr. Robert M. Hoffman, who is also the senior and corresponding author of both publications, describe an “orthotopic” implantation of EwS PDX in the chest wall of nude mice (see Murakami T *et al.* 2016 Oncotarget, PMID 27286459; Miyake K *et al.* 2018 J Cell Biochem, PMID 28681998).

Notably, in a clinical setting, only a minority of primary EwS arise close to the chest wall (~10% of cases) of which many do not directly involve skeletal bones of the thoracic cage. Indeed, ~15% of all EwS cases arise in soft tissue and may occur at any site of the body including the subcutis, which makes the term “orthotopic” in particular difficult to define in this disease, and which is why EwS is often referred to as “bone or soft tissue sarcoma” (for review see Grünewald TG *et al.* 2018 Nature Reviews Disease Primers).

Strikingly, in rare occasions, also molecularly confirmed central nervous system EwS cases that do not involve any bone or soft tissue were described (see Sturm D *et al.* 2016 Cell), underscoring that the term “orthotopic” is still quite vaguely defined in EwS.

Yet, the vast majority of EwS tumors arises within bones (~85% of cases), mostly affecting long bones of the extremities (for review see Grünewald TG *et al.* 2018 Nature Reviews Disease Primers). Therefore, we decided to repeat key experiments using an orthotopic bone injection model, which has been previously described for EwS (see Hauer K *et al.* 2013 Cancer Research; Stewart E *et al.* 2015 Cell Reports; Stewart E *et al.* 2017 Nature).

These orthotopic bone injection experiments fully confirmed our previous observations made in subcutaneous injection models (see below). Given this fact, and given that we consider bone injection models as being more representative for the majority of EwS cases (~85%), we preferred to not carry out chest wall implantation models as suggested by this Reviewer.

However, following the 3R principles (reduction, refinement, replacement) and because these orthotopic bone injection models are associated – despite stringent analgesia – with a moderate

burden for the mice, we also employed advanced mineralized 3D *in vitro* bone models including human osteoblasts to further confirm the conclusions of our paper. In addition, we used material from eight already established EwS PDX tumors to confirm the link between SOX6, proliferation and tumor growth.

Specifically, we now show the following:

In compliance with the 3R principles, we repeated our *SOX6* knockdown experiments in an advanced mineralized 3D *in vitro* bone model that was built from a non-viable matrix and vital human osteoblasts. In these experiments, we found that silencing of *SOX6* significantly decreased 3D growth of EwS cells, which was accompanied by reduced Ki67 expression. These new results are now reported in the revised Results section.

Page 6, paragraph 2: “Similar to our previous *in vitro* assays, knockdown of *SOX6* reduced growth of EwS cells in this 3D bone model (**Supplementary Fig. 3e**). Interestingly, this effect was accompanied by a reduced expression of the proliferation marker Ki67 (**Supplementary Fig. 3e**), which further supported the involvement of SOX6 in cell cycle progression.”

In addition, we repeated our drug-response assays using Elesclomol in this 3D model, which fully confirmed that Elesclomol treatment reduces growth and induces apoptosis of EwS cells grown in a mineralized bone matrix containing viable human osteoblasts. These new results are now reported in the revised Results section.

Page 8, paragraph 1: “Similar to these *in vivo* models, Elesclomol treatment strongly reduced growth of EwS cells compared to DMSO controls in a 3D *in vitro* mineralized bone model, and strongly induced apoptosis as evidenced by staining for cleaved caspase 3 (**Fig. 4k**).”

To further assess the role of SOX6 in proliferation of EwS cells, we took advantage of eight established EwS PDX models. In this set of PDX models, we correlated the intratumoral SOX6 expression levels with the growth kinetics (as measured in at least duplicate mice) of these PDX tumors. These analyses uncovered a significant positive correlation between SOX6 protein expression and speed of tumor growth ($P=0.013$) as well as Ki67 expression levels ($P\leq 0.007$). These new results are now reported in the revised Results section.

Page 6, paragraph 4: “Notably, SOX6 protein expression was significantly positively correlated with the speed of tumor growth and Ki67 immunoreactivity in EwS patient-derived xenograft (PDX) models (**Fig. 3h**), in which we observed a broad range of SOX6 expression levels similar to that in primary EwS tumors (**Fig. 1a, b**).

Since EwS most commonly arises in bone (~85% of cases), we further validated the link between SOX6 expression and EwS growth in an orthotopic bone injection xenotransplantation model, which we have used in the past and which is well-established in the literature (Hauer K *et al.* 2013 Cancer Research; Stewart E *et al.* 2015 Cell Reports; Stewart E *et al.* 2017 Nature).

Strikingly, these experiments demonstrated that knockdown of SOX6 completely abolished growth of EwS xenografts in the bone environment. These new results are now displayed in the new Fig. 3i and reported in the revised Results section.

Page 6, paragraph 4: “In line with this finding, *SOX6* knockdown completely abolished the growth of TC-32/TR/shSOX6 EwS cells in an orthotopic tibial bone injection model *in vivo* (Fig. 3i).”

Reviewers' comments:

Reviewer #1 (Remarks to the Author):

This is a revised manuscript by Marchetto et al. that explores the activation of SOX6 by the EWSR1-FLI1 fusion oncoprotein in Ewing sarcoma thereby activating an oxidative stress pathway that then makes the cells sensitive to Elesclomol. The revision is responsive to the reviewers' comments and adds new data, including the evaluation of additional cell lines and 3D and PDX models. The additional data enhance the association between SOX6 levels and sensitivity to SOX6 silencing and Elesclomol sensitivity. Overall, this manuscript offers a meaningful association between SOX6 expression and possible sensitivity to oxidative stress in Ewing sarcoma. This is an interesting study that should be published.

There are a few additional points that would enhance the manuscript.

1. Since the premise is that high SOX6 enhances the response to Elesclomol, in a reciprocal fashion to the silencing of SOX6 in the SOX6-high Ewing sarcoma cells, the authors could enforce expression of SOX6 in those Ewing sarcoma cell lines with low SOX6 to evaluate whether this increases TXNIP and results in enhanced treatment sensitivity. These data would more strongly support the proposed model shown in Fig. 6i.
2. In addition to the SAOS-2 and MSC-52 cells shown in Fig. 4c, it would be informative to include some of the low SOX6 Ewing sarcoma cells. Do these demonstrate intermediate sensitivity?
3. The plots shown in Fig. 4f and in multiple panels in Figs. 5 and 6 demonstrate the control treatment without variation, likely normalizing them to some standard value. This implies that there was no variability in these experiments, which seems unlikely. It would be preferable to show the actual data so that the experimental variation in both control and treatment conditions can be evaluated. Plots, such as that shown in Fig. 4e demonstrate values relative to one set without normalizing/eliminating variation.

Reviewer #2 (Remarks to the Author):

Marchetto et al. have done a thorough job of addressing my previous review. New experimental data and altered text/discussions support the conclusions, and reasonable rebuttal to other points has been provided. I have no further concerns with the manuscript and support acceptance of the revised submission for publication.

Reviewer #4 (Remarks to the Author):

This reviewer did not review the initial submission, but instead was asked to review the revised manuscript. As such, this reviewer had access to the prior reviews and the authors' responses and revisions to those reviews.

In this manuscript, Marchetto et al. identify SOX6 as a potent transcriptional regulator in various Ewing sarcoma (EWS), and how higher levels of expression in certain EWS cell lines can be exploited as a potential therapeutic approach upon treatment with the oxidative stress-inducing drug, Elesclomol. The authors focus on two EWS cell lines (RDES and TC-32) that have high levels of SOX6 expression, in which they engineered an inducible knockdown of SOX6. The authors show convincing in vitro and in vivo data that SOX6 is important for sustaining proliferation and oncogenic potential in these cell lines. The authors also show that modulation of the sensitivity to Elesclomol is based on expression of SOX6 and upon knockdown of SOX6 significantly decreases

the sensitivity of RDES and TC-32 to Elesclomol. Similarly, the sensitivity of these EwS cell lines to Elesclomol can be reduced by providing antioxidant molecules, Nac or Tiron. Lastly the regulation of TXNIP by SOX6 was evaluated as playing a potential role in the toxicity of Elesclomol in EwS cell lines which is seemingly convincing based on their 'TXNIP knockdown-rescue' experiment in which recovered TXNIP expression sensitizes TC-32 cells to Elesclomol.

Overall, this reviewer found the manuscript to be well-written and convincing. The experiments are well-controlled and generally robust. The authors responded to all of the prior reviewers' concerns, and in the opinion of this reviewer, the additional experiments performed provide additional strength to the results and interpretation of the data. While the reviewers did not agree with every one of the prior reviewers' comments, their arguments were reasonable and convincing. It is this reviewer's opinion that the manuscript meets the bar for publication.

Signed, Stephen Lessnick, MD, PhD

Step-by-step responses to the Reviewers

Authors: We thank all Reviewers for their time spent with our manuscript and their highly valuable and fair comments, which have been addressed in full.

Reviewer #1 (Remarks to the Author):

This is a revised manuscript by Marchetto et al. that explores the activation of SOX6 by the EWSR1-FLI1 fusion oncoprotein in Ewing sarcoma thereby activating an oxidative stress pathway that then makes the cells sensitive to Elesclomol. The revision is responsive to the reviewers' comments and adds new data, including the evaluation of additional cell lines and 3D and PDX models. The additional data enhance the association between SOX6 levels and sensitivity to SOX6 silencing and Elesclomol sensitivity. Overall, this manuscript offers a meaningful association between SOX6 expression and possible sensitivity to oxidative stress in Ewing sarcoma. This is an interesting study that should be published.

Authors: We thank this Reviewer for pointing out that the previous revisions have enhanced our manuscript and that it should be published.

There are a few additional points that would enhance the manuscript.

Authors: We thank this Reviewer for these additional points that we have all addressed.

1. Since the premise is that high SOX6 enhances the response to Elesclomol, in a reciprocal fashion to the silencing of SOX6 in the SOX6-high Ewing sarcoma cells, the authors could enforce expression of SOX6 in those Ewing sarcoma cell lines with low SOX6 to evaluate whether this increases TXNIP and results in enhanced treatment sensitivity. These data would more strongly support the proposed model shown in Fig. 6i.

Authors: We thank this Reviewer for this important remark. To address this question, we infected the *SOX6*-low Ewing sarcoma cell line A673 with a lentivirus harboring an all-in-one doxycycline (dox)-inducible overexpression system containing full-length cDNA of *SOX6*. Indeed, dox-inducible overexpression of *SOX6* in A673 cells was accompanied by a significant induction of *TXNIP* expression and a higher sensitivity toward Elesclomol ($P < 0.01$). These new data are now presented in the new Supplementary Figures 5b, c and presented in the revised Results section:

Page 9, paragraph 6: “Of note, inducible overexpression of *SOX6* in the *SOX6*-low EwS cell line A673 was accompanied by a significant induction of *TXNIP* expression and increased sensitivity toward Elesclomol ($P < 0.01$) (Supplementary Fig. 5b, c).”

2. In addition to the SAOS-2 and MSC-52 cells shown in Fig. 4c, it would be informative to include some of the low *SOX6* Ewing sarcoma cells. Do these demonstrate intermediate sensitivity?

Authors: We thank this Reviewer for this additional question, and fully agree that it would be informative to include a *SOX6*-low EwS cell line. To address this question, we analyzed the Elesclomol sensitivity of the *SOX6*-low EwS cell line A673. As shown in the revised Figs. 4c–e, this cell line exhibits similarly low *SOX6* levels than the osteosarcoma cell line SAOS-2 and the mesenchymal stem cell line MSC-52, which is associated with a similarly low Elesclomol sensitivity. These new results are now described in the revised Results section:

Page 7, paragraph 4: “Indeed, in validation drug-response assays, Elesclomol strongly decreased viability of EwS cells with high *SOX6* levels while the osteosarcoma cell line SAOS-2 and non-transformed human primary MSC line MSC-52 that exhibit low *SOX6* expression levels were relatively resistant (**Fig. 4c, d**). Likewise, the *SOX6*-low EwS cell line A673 exhibited a similarly low sensitivity toward Elesclomol than SAOS-2 and MSC-52 (**Fig. 4c, d**). The high sensitivity of *SOX6*-high EwS cells toward Elesclomol appeared to be independent of proliferation under normal conditions, since the osteosarcoma cell line SAOS-2 proliferated even more than the tested *SOX6*-high EwS cells (**Fig. 4e**).”

3. The plots shown in Fig. 4f and in multiple panels in Figs. 5 and 6 demonstrate the control treatment without variation, likely normalizing them to some standard value. This implies that there was no variability in these experiments, which seems unlikely. It would be preferable to show the actual data so that the experimental variation in both control and treatment conditions can be evaluated. Plots, such as that shown in Fig. 4e demonstrate values relative to one set without normalizing/eliminating variation.

Authors: We thank this Reviewer for this remark and fully agree that it is preferable to display the variability of the controls whenever technically possible. Accordingly, we now display the data shown in Fig. 4f relative to one set as done in Fig. 4e. Similarly, we have adapted the display of the data concerning mRNA expression analyses and Elesclomol drug-response assays of Figs. 6b, d, g, and h.

However, for assays using the very sensitive fluorochromes MitoSOX Red and DCF-DA to study very labile reactive oxygen species (ROS) and oxidative stress, this was not possible. While we noted very robust fold changes in the study groups across experiments and although the technical replicates in each experiment showed only little variation, we observed variation across biologically independent experiments in the raw values making it necessary to normalize the data on the actual control (set as 100%). It is well-known that minor variations in light exposure, ambient temperature, and incubation times can have strong effects on these sensitive fluorochromes. Hence, it is a standard procedure for these ROS-assays to set for each experiment

the controls as 1 (or 100%) and to only compare the actual fold changes across experiments (for comparison see: Quintana-Cabrera R *et al.* 2012 Nature Communications; Lückel C *et al.* 2019 Nature Communications; Wang P *et al.* 2019 Nature Communications; Zhang W *et al.* 2019 Nature Communications). Hence, we would prefer to display the data of these ROS-assays in Figs. 5 and 6 as normalized fold changes.

However, to address this Reviewer's recommendation, we now explained the way of data normalization for these ROS-assays in the revised Methods section and refer to the corresponding references:

Page 21, paragraph 3: "To account for variations in relative fluorescence of MitoSOX Red and DCF-DA across experiments, data are shown as fold changes of the experimental groups over the respective controls that were normalized to 100% in each experiment as described previously (Quintana-Cabrera R *et al.* 2012 Nature Communications; Lückel C *et al.* 2019 Nature Communications; Wang P *et al.* 2019 Nature Communications; Zhang W *et al.* 2019 Nature Communications)."

Reviewer #2 (Remarks to the Author):

Marchetto et al. have done a thorough job of addressing my previous review. New experimental data and altered text/discussions support the conclusions, and reasonable rebuttal to other points has been provided. I have no further concerns with the manuscript and support acceptance of the revised submission for publication.

Authors: We thank this Reviewer for the very positive evaluation of the revised version of our manuscript and for pointing out that it should be accepted for publication.

Reviewer #4 (Remarks to the Author):

This reviewer did not review the initial submission, but instead was asked to review the revised manuscript. As such, this reviewer had access to the prior reviews and the authors' responses and revisions to those reviews.

In this manuscript, Marchetto et al. identify SOX6 as a potent transcriptional regulator in various Ewing sarcoma (EwS), and how higher levels of expression in certain EwS cell lines can be exploited as a potential therapeutic approach upon treatment with the oxidative stress-inducing drug, Elesclomol. The authors focus on two EwS cell lines (RDES and TC-32) that have high levels of SOX6 expression, in which they engineered an inducible knockdown of SOX6. The authors show convincing *in vitro* and *in vivo* data that SOX6 is important for sustaining proliferation and oncogenic potential in these cell lines. The authors also show that modulation of the sensitivity to Elesclomol is based on expression of SOX6 and upon knockdown of SOX6 significantly decreases the sensitivity of RDES and TC-32 to Elesclomol. Similarly, the

sensitivity of these EwS cell lines to Elesclomol can be reduced by providing antioxidants molecules, Nac or Tiron. Lastly the regulation of TXNIP by SOX6 was evaluated as playing a potential role in the toxicity of Elesclomol in EwS cell lines which is seemingly convincing based on their 'TXNIP knockdown-rescue' experiment in which recovered TXNIP expression sensitizes TC-32 cells to Elesclomol.

Overall, this reviewer found the manuscript to be well-written and convincing. The experiments are well-controlled and generally robust. The authors responded to all of the prior reviewers' concerns, and in the opinion on this reviewer, the additional experiments performed provide additional strength to the results and interpretation of the data. While the reviewers did not agree with every one of the prior reviewers' comments, their arguments were reasonable and convincing. It is this reviewer's opinion that manuscript meets the bar for publication.

Signed, Stephen Lessnick, MD, PhD

Authors: We thank Dr. Lessnick for his very positive evaluation of the revised version of our manuscript and for highlighting its scientific quality and robustness. We fully agree with Dr. Lessnick that we have responded to all prior Reviewers' concerns, and that our additional experiments have further strengthened our paper. We thank Dr. Lessnick for pointing out that our revised manuscript meets the criteria for publication.

REVIEWERS' COMMENTS:

Reviewer #1 (Remarks to the Author):

The authors have carefully and thoroughly responded to the concerns raised during review, and I consider the manuscript appropriate for publication.

Step-by-step responses to the Reviewer

Reviewer #1 (Remarks to the Author):

The authors have carefully and thoroughly responded to the concerns raised during review, and I consider the manuscript appropriate for publication.

Authors: We thank the Reviewer for the time spent with our manuscript and for the positive evaluation of our revisions.